# Pure Message Passing Can Estimate Common Neighbor for Link Prediction

## Abstract

Message Passing Neural Networks (MPNNs) have emerged as the *de facto* standard in graph representation learning. However, when it comes to link prediction, they are not always superior to simple heuristics such as Common Neighbor (CN). This discrepancy stems from a fundamental limitation: while MPNNs excel in node-level representation, they stumble with encoding the joint structural features essential to link prediction, like CN. To bridge this gap, we posit that, by harnessing the orthogonality of input vectors, pure message-passing can indeed capture joint structural features. Specifically, we study the proficiency of MPNNs in approximating CN heuristics. Based on our findings, we introduce the Message Passing Link Predictor (MPLP), a novel link prediction model. MPLP taps into quasi-orthogonal vectors to estimate link-level structural features, all while preserving the node-level complexities. Moreover, our approach demonstrates that leveraging message-passing to capture structural features could offset MPNNs' expressiveness limitations at the expense of estimation variance. We conduct experiments on benchmark datasets from various domains, where our method consistently outperforms the baseline methods.

## 1 Introduction

Link prediction is a cornerstone task in the field of graph machine learning, with broad-ranging implications across numerous industrial applications. From identifying potential new acquaintances on social networks (Liben-Nowell & Kleinberg, 2003) to predicting protein interactions (Szklarczyk et al., 2019), from enhancing recommendation systems (Koren et al., 2009) to completing knowledge graphs (Zhu et al., 2021), the impact of link prediction is felt across diverse domains. Recently, with the advent of Graph Neural Networks (GNNs) (Kipf & Welling, 2017) and more specifically, Message-Passing Neural Networks (MPNNs) (Gilmer et al., 2017), these models have become the primary tools for tackling link prediction tasks. Despite the resounding success of MPNNs in the realm of node and graph classification tasks (Kipf & Welling, 2017; Hamilton et al., 2018; Veličković et al., 2018; Xu et al., 2018), it is intriguing to note that their performance in link prediction does not always surpass that of simpler heuristic methods (Hu et al., 2021).

Zhang et al. (2021) highlights the limitations of GNNs/MPNNs for link prediction tasks arising from its intrinsic property of permutation invariance. Owing to this property, isomorphic nodes invariably receive identical representations. This poses a challenge when attempting to distinguish links whose endpoints are isomorphic nodes. As illustrated in Figure 1a, nodes $v_1$ and $v_3$ share a Common Neighbor $v_2$, while nodes $v_1$ and $v_5$ do not. Ideally, due to their disparate local structures, these two links $(v_1, v_3)$ and $(v_1, v_5)$ should receive distinct predictions. However, the permutation invariance of MPNNs results in identical representations for nodes $v_3$ and $v_5$, leading to identical predictions for the two links. As Zhang et al. (2021) asserts, such node-level representation, even with the most expressive MPNNs, **cannot** capture structural link representation such as Common Neighbors (CN), a critical aspect of link prediction.

In this work, we posit that the pure Message Passing paradigm (Gilmer et al., 2017) can indeed capture structural link representation by exploiting orthogonality within the vector space. We begin by presenting a motivating example, considering a non-attributed graph as depicted in Figure 1a. In order to fulfill the Message Passing's requirement for node vectors as input, we assign a one-hot vector to each node $v_i$, such that the $i$-th dimension has a value of one, with the rest set to zero.

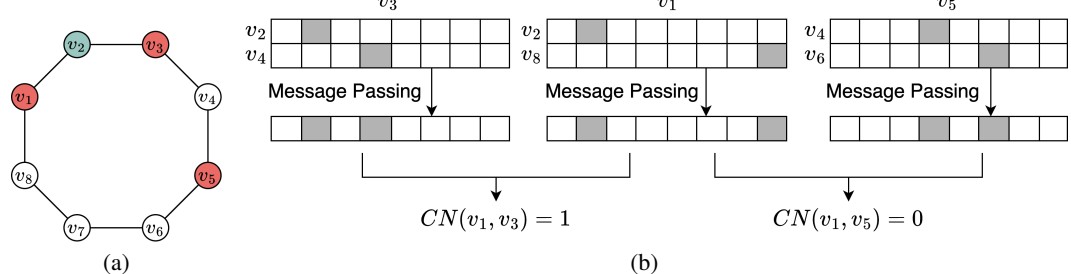

Figure 1: (a) Isomorphic nodes result in identical MPNN node representation, making it impossible to distinguish links such as $(v_1, v_3)$ and $(v_1, v_5)$ based on these representations. (b) MPNN counts Common Neighbor through the inner product of neighboring nodes' one-hot representation.

These vectors, viewed as *signatures* rather than mere permutation-invariant node representations, can illuminate pairwise relationships. Subsequently, we execute a single iteration of message passing as shown in Figure 1b, updating each node's vector by summing the vector of its neighbors. This process enables us to compute CN for any node pair by taking the inner product of the vectors of the two target nodes.

At its core, this naive method employs an orthonormal basis as the node signatures, thereby ensuring that the inner product of distinct nodes' signatures is consistently zero. While this approach effectively computes CN, its scalability poses a significant challenge, given that its space complexity is quadratically proportional to the size of the graph. To overcome this, we draw inspiration from DotHash (Nunes et al., 2023) and capitalize on the premise that the family of vectors almost orthogonal to each other swells exponentially, even with just linearly scaled dimensions Kainen & Kůrková (1993). Instead of relying on the orthogonal basis, we can propagate these quasi-orthogonal (QO) vectors and utilize the inner product to estimate the joint structural information of any node pair. Furthermore, by strategically selecting which pair of node signatures to compute the inner product, we can boost the expressiveness of MPNNs to estimate substructures—a feat previously deemed impossible in the literature (Chen et al., 2020).

In sum, our paper presents several pioneering advances in the realm of GNNs for link prediction:

- We are the first, both empirically and theoretically, to delve into the proficiency of GNNs in approximating heuristic predictors like CN for link prediction. This uncovers a previously uncharted territory in GNN research.
- Drawing upon the insights gleaned from GNNs' capabilities in counting CN, we introduce **MPLP**, a novel link prediction model. Uniquely, MPLP discerns joint structures of links and their associated substructures within a graph, setting a new paradigm in the field.
- Our empirical investigations provide compelling evidence of MPLP's dominance. Benchmark tests reveal that MPLP not only holds its own but outstrips state-of-the-art models in link prediction performance.

## 2 PRELIMINARIES AND RELATED WORK

**Notations.** Consider an undirected graph $G = (V, E, \boldsymbol{X})$, where $V$ represents the set of nodes with cardinality $n$, indexed as $\{1, \ldots, n\}$, $E \subseteq V \times V$ denotes the observed set of edges, and $\boldsymbol{X}_{i,:} \in \mathbb{R}^{F_x}$ encapsulates the attributes associated with node $i$. Additionally, let $\mathcal{N}_v$ signify the neighborhood of a node $v$, that is $\mathcal{N}_v = \{u | \text{SPD}(u, v) = 1\}$ where the function $\text{SPD}(\cdot, \cdot)$ measures the shortest path distance between two nodes. Furthermore, the node degree of $v$ is given by $d_v = |\mathcal{N}_v|$. To generalize, we introduce the shortest path neighborhood $\mathcal{N}_v^s$, representing the set of nodes that are $s$ hops away from node $v$, defined as $\mathcal{N}_v^s = \{u | \text{SPD}(u, v) = s\}$.

**Link predictions.** Alongside the observed set of edges $E$, there exists an unobserved set of edges, which we denote as $E_c \subseteq V \times V \setminus E$. This unobserved set encompasses edges that are either absent from the original observation or are anticipated to materialize in the future within the graph $G$. Consequently, we can formulate the link prediction task as discerning the unobserved set of edges $E_c$. Heuristics link predictors include Common Neighbor (CN) (Liben-Nowell & Kleinberg, 2003), Adamic-Adar index (AA) (Adamic & Adar, 2003), and Resource Allocation (RA) (Zhou

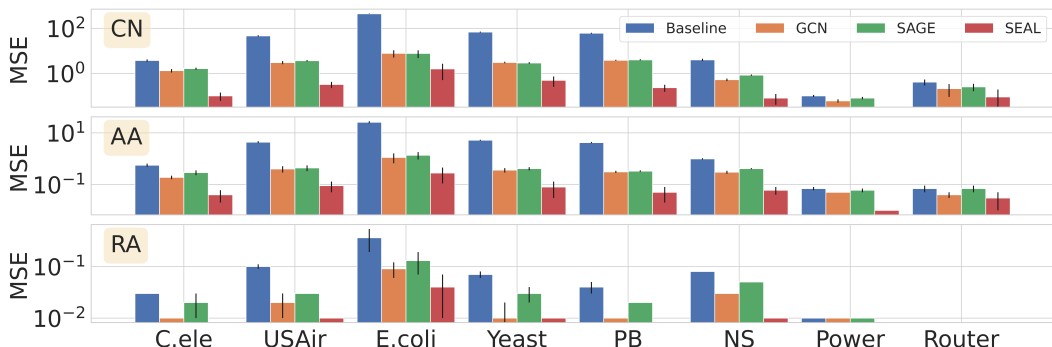

Figure 2: GNNs estimate CN, AA and RA via MSE regression, using the mean value as a Baseline. Lower values are better.

et al., 2009). CN is simply counting the cardinality of the common neighbors, while AA and RA count them weighted to reflect their relative importance as a common neighbor.

$$\text{CN}(u,v) = \sum_{k \in \mathcal{N}_u \bigcap \mathcal{N}_v} 1 \;\; ; \;\; \text{AA}(u,v) = \sum_{k \in \mathcal{N}_u \bigcap \mathcal{N}_v} \frac{1}{\log d_k} \;\; ; \;\; \text{RA}(u,v) = \sum_{k \in \mathcal{N}_u \bigcap \mathcal{N}_v} \frac{1}{d_k}. \quad (1)$$

Though heuristic link predictors are effective across various graph domains, their growing computational demands clash with the need for low latency. To mitigate this, approaches like ELPH (Chamberlain et al., 2022) and DotHash (Nunes et al., 2023) propose using estimations rather than exact calculations for these predictors. Our study, inspired by these works, seeks to further refine techniques for efficient link predictions. A detailed comparison with related works and our method is available in Appendix A.

**GNNs for link prediction.** The advent of graphs incorporating node attributes has caused a significant shift in research focus toward methods grounded in GNNs. Most practical GNNs follow the paradigm of the Message Passing (Gilmer et al., 2017). It can be formulated as:

$$\boldsymbol{h}_v^{(l+1)} = \text{UPDATE}\left(\{\boldsymbol{h}_v^{(l)}, \text{AGGREGATE}\left(\{\boldsymbol{h}_u^{(l)}, \boldsymbol{h}_v^{(l)}, \forall u \in \mathcal{N}_v\}\right)\}\right), \quad (2)$$

where $\boldsymbol{h}_v^{(l)}$ represents the vector of node $v$ at layer $l$ and $\boldsymbol{h}_v^{(0)} = \boldsymbol{X}_{v,:}$. For simplicity, we use $\boldsymbol{h}_v$ to represent the node vector at the last layer. The specific choice of the neighborhood aggregation function, AGGREGATE($\cdot$), and the updating function, UPDATE($\cdot$), dictates the instantiation of the GNN model, with different choices leading to variations of model architectures. In the context of link prediction tasks, the GAE model (Kipf & Welling, 2016) derives link representation, $\boldsymbol{h}(i,j)$, as a Hadamard product of the target node pair representations, $\boldsymbol{h}_{(i,j)} = \boldsymbol{h}_i \odot \boldsymbol{h}_j$. Despite its seminal approach, the SEAL model (Zhang & Chen, 2018), which labels nodes based on proximity to target links and then performs message-passing for each target link, is hindered by computational expense, limiting its scalability. Efficient alternatives like ELPH (Chamberlain et al., 2022) estimate node labels, while NCNC (Wang et al., 2023) directly learns edgewise features by aggregating node representations of common neighbors.

## 3 CAN MESSAGE PASSING COUNT COMMON NEIGHBOR?

In this section, we delve deep into the potential of MPNNs for heuristic link predictor estimation. We commence with an empirical evaluation to recognize the proficiency of MPNNs in approximating link predictors. Following this, we unravel the intrinsic characteristics of 1-layer MPNNs, shedding light on their propensity to act as biased estimators for heuristic link predictors and proposing an unbiased alternative. Ultimately, we cast light on how successive rounds of message passing can estimate the number of walks connecting a target node pair with other nodes in the graph. All proofs related to the theorem are provided in Appendix E.

### 3.1 ESTIMATION VIA MEAN SQUARED ERROR REGRESSION

To explore the capacity of MPNNs in capturing the overlap information inherent in heuristic link predictors, such as CN, AA and RA, we conduct an empirical investigation, adopting the GAE

framework (Kipf & Welling, 2016) with GCN (Kipf & Welling, 2017) and SAGE (Hamilton et al., 2018) as representative encoders. SEAL (Zhang & Chen, 2018), known for its proven proficiency in capturing heuristic link predictors, serves as a benchmark in our comparison. Additionally, we select a non-informative baseline estimation, simply using the mean of the heuristic link predictors on the training sets. The datasets comprise eight non-attributed graphs (more details in Section 5). Given that GNN encoders require node features for initial representation, we have to generate such features for our non-attributed graphs. We achieved this by sampling from a high-dimensional Gaussian distribution with a mean of $0$ and standard deviation of $1$. Although one-hot encoding is frequently employed for feature initialization on non-attributed graphs, we choose to forgo this approach due to the associated time and space complexity.

To evaluate the ability of GNNs to estimate CN information, we adopt a training procedure analogous to a conventional link prediction task. However, we reframe the task as a regression problem aimed at predicting heuristic link predictors, rather than a binary classification problem predicting link existence. This shift requires changing the objective function from cross-entropy to Mean Squared Error (MSE). Such an approach allows us to directly observe GNNs' capacity to approximate heuristic link predictors.

Our experimental findings, depicted in Figure 2, reveal that GCN and SAGE both display an ability to estimate heuristic link predictors, albeit to varying degrees, in contrast to the non-informative baseline estimation. More specifically, GCN demonstrates a pronounced aptitude for estimating RA and nearly matches the performance of SEAL on datasets such as C.ele, Yeast, and PB. Nonetheless, both GCN and SAGE substantially lag behind SEAL in approximating CN and AA. In the subsequent section, we delve deeper into the elements within the GNN models that facilitate this approximation of link predictors while also identifying factors that impede their accuracy.

## 3.2 Estimation capabilities of GNNs for link predictors

GNNs exhibit the capability of estimating link predictors. In this section, we aim to uncover the mechanisms behind these estimations, hoping to offer insights that could guide the development of more precise and efficient methods for link prediction. We commence with the following theorem:

**Theorem 1.** *Let $G = (V, E)$ be a non-attributed graph and consider a 1-layer GCN/SAGE. Define the input vectors $\boldsymbol{X} \in \mathbb{R}^{N \times F}$ initialized randomly from a zero-mean distribution with standard deviation $\sigma_{node}$. Additionally, let the weight matrix $\boldsymbol{W} \in \mathbb{R}^{F' \times F}$ be initialized from a zero-mean distribution with standard deviation $\sigma_{weight}$. After performing message passing, for any pair of nodes $\{(u, v) | (u, v) \in V \times V \setminus E\}$, the expected value of their inner product is given by:*

$$\text{GCN: } \mathbb{E}(\boldsymbol{h}_u \cdot \boldsymbol{h}_v) = \frac{C}{\sqrt{\hat{d}_u \hat{d}_v}} \sum_{k \in \mathcal{N}_u \bigcap \mathcal{N}_v} \frac{1}{\hat{d}_k} \text{ ; } \text{SAGE: } \mathbb{E}(\boldsymbol{h}_u \cdot \boldsymbol{h}_v) = \frac{C}{\sqrt{d_u d_v}} \sum_{k \in \mathcal{N}_u \bigcap \mathcal{N}_v} 1,$$

*where $\hat{d}_v = d_v + 1$ and the constant $C$ is defined as $C = \sigma_{node}^2 \sigma_{weight}^2 F F'$.*

The theorem suggests that given proper initialization of input vectors and weight matrices, MPNN-based models, such as GCN and SAGE, can adeptly approximate heuristic link predictors. This makes them apt for encapsulating joint structural features of any node pair. Interestingly, SAGE predominantly functions as a CN estimator, whereas the aggregation function in GCN grants it the ability to weigh the count of common neighbors in a way similar to RA. This particular trait of GCN is evidenced by its enhanced approximation of RA, as depicted in Figure 2.

**Quasi-orthogonal vectors.** The GNN's capability to approximate heuristic link predictors is primarily grounded in the properties of their input vectors in a linear space. When vectors are sampled from a high-dimensional linear space, they tend to be quasi-orthogonal, implying that their inner product is nearly 0 w.h.p. With message-passing, these QO vectors propagate through the graph, yielding in a linear combination of QO vectors at each node. The inner product between pairs of QO vector sets essentially echoes the norms of shared vectors while nullifying the rest. Such a trait enables GNNs to estimate CN through message-passing. A key advantage of QO vectors, especially when compared with orthonormal basis, is their computational efficiency. For a modest linear increment in space dimensions, the number of QO vectors can grow exponentially, given an acceptable margin of error (Kainen & Kůrková, 1993). An intriguing observation is that the orthogonality of QO vectors remains intact even after GNNs undergo linear transformations post message-passing,

attributed to the randomized weight matrix initialization. This mirrors the dimension reduction observed in random projection (Johnson & Lindenstrauss, 1984).

**Limitations.** While GNNs manifest a marked ability in estimating heuristic link predictors, they are not unbiased estimators and can be influenced by factors such as node pair degrees, thereby compromising their accuracy. Another challenge when employing such MPNNs is their limited generalization to unseen nodes. The neural networks, exposed to randomly generated vectors, may struggle to transform newly added nodes in the graph with novel random vectors. This practice also violates the permutation-invariance principle of GNNs when utilizing random vectors as node representation. It could strengthen generalizability if we regard these randomly generated vectors as signatures of the nodes, instead of their node features, and circumvent the use of MLPs for them.

**Unbiased estimator.** Addressing the biased element in Theorem 1, we propose the subsequent instantiation for the message-passing functions:

$$\boldsymbol{h}_v^{(l+1)} = \sum_{u \in \mathcal{N}_v} \boldsymbol{h}_u^{(l)}. \tag{3}$$

Such an implementation aligns with the SAGE model that employs sum aggregation devoid of self-node propagation. This methodology also finds mention in DotHash (Nunes et al., 2023), serving as a cornerstone for our research. With this kind of message-passing design, the inner product of any node pair signatures can estimate CN impartially:

**Theorem 2.** *Let $G = (V, E)$ be a graph, and let the vector dimension be given by $F \in \mathbb{N}_+$. Define the input vectors $\boldsymbol{X} = (X_{i,j})$, which are initialized from a random variable* x *having a mean of $0$ and a standard deviation of $\frac{1}{\sqrt{F}}$. Using the 1-layer message-passing in Equation 3, for any pair of nodes $\{(u,v)|(u,v) \in V \times V\}$, the expected value and variance of their inner product are:*

$$\mathbb{E}(\boldsymbol{h}_u \cdot \boldsymbol{h}_v) = \mathrm{CN}(u,v),$$

$$\mathrm{Var}(\boldsymbol{h}_u \cdot \boldsymbol{h}_v) = \frac{1}{F}\left(d_u d_v + \mathrm{CN}(u,v)^2 - 2\mathrm{CN}(u,v)\right) + F\mathrm{Var}(\mathrm{x}^2)\mathrm{CN}(u,v).$$

Though this estimator provides an unbiased estimate for CN, its accuracy can be affected by its variance. Specifically, DotHash recommends selecting a distribution for input vector sampling from vertices of a hypercube with unit length, which curtails variance given that $\mathrm{Var}(\mathrm{x}^2) = 0$. However, the variance influenced by the graph structure isn't adequately addressed, and this issue will be delved into in Section 4.

**Orthogonal node attributes.** Both Theorem 1 and Theorem 2 underscore the significance of quasi orthogonality in input vectors, enabling message-passing to efficiently count CN. Intriguingly, in most attributed graphs, node attributes, often represented as bag-of-words (Purchase et al., 2022), exhibit inherent orthogonality. This brings forth a critical question: In the context of link prediction, do GNNs primarily approximate neighborhood overlap, sidelining the intrinsic value of node attributes? We earmark this pivotal question for in-depth empirical exploration in Appendix C, where we find that random vectors as input to GNNs can catch up with or even outperform node attributes.

### 3.3 MULTI-LAYER MESSAGE PASSING

Theorem 2 elucidates the estimation of CN based on a single iteration of message passing. This section explores the implications of multiple message-passing iterations and the properties inherent to the iteratively updated node signatures. We begin with a theorem delineating the expected value of the inner product for two nodes' signatures derived from any iteration of message passing:

**Theorem 3.** *Under the conditions defined in Theorem 2, let $\boldsymbol{h}_u^{(l)}$ denote the vector for node $u$ after the $l$-th message-passing iteration. We have:*

$$\mathbb{E}\left(\boldsymbol{h}_u^{(p)} \cdot \boldsymbol{h}_v^{(q)}\right) = \sum_{k \in V} |walks^{(p)}(k,u)||walks^{(q)}(k,v)|,$$

*where $|walks^{(l)}(u,v)|$ counts the number of length-l walks between nodes $u$ and $v$.*

This theorem posits that the message-passing procedure computes the number of walks between the target node pair and all other nodes. In essence, each message-passing trajectory mirrors the path

of the corresponding walk. As such, $\boldsymbol{h}_u^{(l)}$ aggregates the initial QO vectors originating from nodes reachable by length-$l$ walks from node $u$. In instances where multiple length-$l$ walks connect node $k$ to $u$, the associated QO vector $\boldsymbol{X}_{k,:}$ is incorporated into the sum $|\text{walks}^{(l)}(k, u)|$ times.

One might surmise a paradox, given that message-passing calculates the number of walks, not nodes. However, in a simple graph devoid of self-loops, where at most one edge can connect any two nodes, it is guaranteed that $|\text{walks}^{(1)}(u, v)| = 1$ iff $\text{SPD}(u, v) = 1$. Consequently, the quantity of length-1 walks to a target node pair equates to CN, a first-order heuristic. It's essential to recognize, however, that $|\text{walks}^{(l)}(u, v)| \geq 1$ only implies $\text{SPD}(u, v) \leq l$. This understanding becomes vital when employing message-passing for estimating the local structure of a target node pair in Section 4.

## 4 METHOD

In this section, we introduce our novel link prediction model, denoted as **MPLP**. Distinctively designed, MPLP leverages the pure essence of the message-passing mechanism to adeptly learn structural information. Not only does MPLP encapsulate the local structure of the target node pair by assessing node counts based on varying shortest-path distances, but it also pioneers in estimating the count of triangles linked to any of the target node pair— an ability traditionally deemed unattainable for GNNs (Chen et al., 2020).

**Node representation.** While MPLP is specifically designed for its exceptional structural capture, it also embraces the inherent attribute associations of graphs that speak volumes about individual node characteristics. To fuse the attributes (if they exist in the graph) and structures, MPLP begins with a GNN, utilized to encode node $u$'s representation: $\text{GNN}(u) \in \mathbb{R}^{F_x}$. This node representation will be integrated into the structural features when constructing the QO vectors. Importantly, this encoding remains flexible, permitting the choice of any node-level GNN.

Figure 3: Representation of the target link $(u, v)$ within our model (MPLP), with nodes color-coded based on their distance from the target link.

### 4.1 QO VECTORS CONSTRUCTION

**Probabilistic hypercube sampling.** Though deterministic avenues for QO vector construction are documented (Kainen, 1992; Kainen & Kurkova, 2020), our preference leans toward probabilistic techniques for their inherent simplicity. We inherit the sampling paradigm from DotHash (Nunes et al., 2023), where each node $k$ is assigned with a node signature $\boldsymbol{h}_k^{(0)}$, acquired via random sampling from the vertices of an $F$-dimensional hypercube with unit vector norms. Consequently, the sampling space for $\boldsymbol{h}_k^{(0)}$ becomes $\{-1/\sqrt{F}, 1/\sqrt{F}\}^F$.

**Harnessing One-hot hubs for variance reduction.** The stochastic nature of our estimator brings along an inevitable accompaniment: variance. Theorem 2 elucidates that a graph's topology can augment estimator variance, irrespective of the chosen QO vector distribution. At the heart of this issue is the imperfectness of quasi-orthogonality. While a pair of vectors might approach orthogonality, the same cannot be confidently said for the subspaces spanned by larger sets of QO vectors.

Capitalizing on the empirical observation that real-world graphs predominantly obey the power-law distribution (Barabási & Albert, 1999), we discerned a strategy to control variance. Leveraging the prevalence of high-degree nodes—or *hubs*—we designate unique one-hot vectors for the foremost hubs. Consider the graph's top-$b$ hubs; while other nodes draw their QO vectors from a hypercube $\{-1/\sqrt{F-b}, 1/\sqrt{F-b}\}^{F-b} \bigtimes \{0\}^b$, these hubs are assigned one-hot vectors from $\{0\}^{F-b} \bigtimes \{0, 1\}^b$, reserving a distinct subspace of the linear space to safeguard orthogonality. Note that when new nodes are added to the graph, their QO vectors are sampled the same way as the non-hub nodes, which can ensure a tractable computation complexity.

**Norm rescaling to facilitate weighted counts.** Theorem 1 alludes to an intriguing proposition: the estimator's potential to encapsulate not just CN, but also RA. Essentially, RA and AA are nuanced heuristics translating to weighted enumerations of shared neighbors, based on their node degrees. In Theorem 2, such counts are anchored by vector norms during dot products. MPLP enhances this count methodology by rescaling node vector norms, drawing inspiration from previous works (Nunes et al., 2023; Yun et al., 2021). This rescaling is determined by the node's representation, GNN($u$), and its degree $d_u$. The rescaled vector is formally expressed as:

$$\tilde{\boldsymbol{h}}_k^{(0)} = f(\text{GNN}(k)||[d_k]) \cdot \boldsymbol{h}_k^{(0)}, \tag{4}$$

where $f\colon \mathbb{R}^{F_x+1} \to \mathbb{R}$ is an MLP mapping the node representation and degree to a scalar, enabling the flexible weighted count paradigm.

## 4.2 STRUCTURAL FEATURE ESTIMATIONS

**Node label estimation.** The estimator in Theorem 2 can effectively quantify CN. Nonetheless, solely relying on CN fails to encompass diverse topological structures embedded within the local neighborhood. To offer a richer representation, we turn to Distance Encoding (DE) (Li et al., 2020). DE acts as an adept labeling tool (Zhang et al., 2021), demarcating nodes based on their shortest-path distances relative to a target node pair. For a given pair $(u, v)$, a node $k$ belongs to DE($p, q$) iff SPD($u, k$) $= p$ and SPD($v, k$) $= q$. Unlike its usage as node labels, we opt to enumerate these labels, producing a link feature defined by #($p, q$) $= |\text{DE}(p, q)|$. Our model adopts a philosophy akin to ELPH (Chamberlain et al., 2022), albeit with a distinct node-estimation mechanism.

Returning to Theorem 3, we recall that message-passing as in Equation 3 essentially corresponds to walks. Our ambition to enumerate nodes necessitates a single-layer message-passing alteration, reformulating Equation 3 to:

$$\boldsymbol{\eta}_v^s = \sum_{k \in \mathcal{N}_v^s} \tilde{\boldsymbol{h}}_k^{(0)}. \tag{5}$$

Here, $\mathcal{N}_v^s$ pinpoints $v$'s shortest-path neighborhoods distanced by the shortest-path $s$. This method sidesteps the duplication dilemma highlighted in Theorem 3, ensuring that $\boldsymbol{\eta}_v^s$ aggregates at most one QO vector per node. Similar strategies are explored in (Abboud et al., 2022; Feng et al., 2022).

For a tractable computation, we limit the largest shortest-path distance as $r \geq max(p, q)$. Consequently, to capture the varied proximities of nodes to the target pair $(u, v)$, we can deduce:

$$\#(p, q) = \begin{cases} \mathbb{E}(\boldsymbol{\eta}_u^p \cdot \boldsymbol{\eta}_v^q), & r \geq p, q \geq 1 \\ |\mathcal{N}_v^q| - \sum_{1 \leq s \leq r} \#(s, q), & p = 0 \\ |\mathcal{N}_u^p| - \sum_{1 \leq s \leq r} \#(p, s), & q = 0 \end{cases} \tag{6}$$

Concatenating the resulting estimates yields the expressive structural features of MPLP.

**Shortcut removal.** The intricately designed structural features improve the expressiveness of MPLP. However, this augmented expressiveness introduces susceptibility to distribution shifts during link prediction tasks (Dong et al., 2022). Consider a scenario wherein the neighborhood of a target node pair contains a node $k$. Node $k$ resides a single hop away from one of the target nodes but requires multiple steps to connect with the other. When such a target node pair embodies a positive instance in the training data (indicative of an existing link), node $k$ can exploit both the closer target node and the link between the target nodes as a shortcut to the farther one. This dynamic ensures that for training-set positive instances, the maximum shortest-path distance from any neighboring node to the target pair is constrained to the smaller distance increased by one. This can engender a discrepancy in distributions between training and testing phases, potentially diminishing the model's generalization capability.

To circumvent this pitfall, we adopt an approach similar to preceding works (Zhang & Chen, 2018; Yin et al., 2022; Wang et al., 2023; Jin et al., 2022). Specifically, we exclude target links from the original graph during each training batch, as shown by the dash line in Figure 3. This maneuver ensures these links are not utilized as shortcuts, thereby preserving the fidelity of link feature construction.

Table 1: Link prediction results on non-attributed benchmarks evaluated by Hits@50. The format is average score ± standard deviation. The top three models are colored by **First**, **Second**, **Third**.

| | USAir | NS | PB | Yeast | C.ele | Power | Router | E.coli |
|---|---|---|---|---|---|---|---|---|
| CN | $80.52_{\pm4.07}$ | $74.00_{\pm1.98}$ | $37.22_{\pm3.52}$ | $72.60_{\pm3.85}$ | $47.67_{\pm10.87}$ | $11.57_{\pm0.55}$ | $9.38_{\pm1.05}$ | $51.74_{\pm2.70}$ |
| AA | $85.51_{\pm2.25}$ | $74.00_{\pm1.98}$ | $39.48_{\pm3.53}$ | $73.62_{\pm1.01}$ | $58.34_{\pm2.88}$ | $11.57_{\pm0.55}$ | $9.38_{\pm1.05}$ | $68.13_{\pm1.61}$ |
| RA | $85.95_{\pm1.83}$ | $74.00_{\pm1.98}$ | $38.94_{\pm3.54}$ | $73.62_{\pm1.01}$ | $61.47_{\pm4.59}$ | $11.57_{\pm0.55}$ | $9.38_{\pm1.05}$ | $74.45_{\pm0.55}$ |
| GCN | $73.29_{\pm4.70}$ | $78.32_{\pm2.57}$ | $37.32_{\pm4.69}$ | $73.15_{\pm2.41}$ | $40.68_{\pm5.45}$ | $15.40_{\pm2.90}$ | $24.42_{\pm4.59}$ | $61.02_{\pm11.91}$ |
| SAGE | $83.81_{\pm3.09}$ | $56.62_{\pm9.41}$ | $\textbf{47.26}_{\pm2.53}$ | $71.06_{\pm5.12}$ | $58.97_{\pm4.77}$ | $6.89_{\pm0.95}$ | $42.25_{\pm4.32}$ | $75.60_{\pm2.40}$ |
| SEAL | $\textbf{90.47}_{\pm3.00}$ | $\textbf{86.59}_{\pm3.03}$ | $44.47_{\pm2.86}$ | $\textbf{83.92}_{\pm1.17}$ | $\textbf{64.80}_{\pm4.23}$ | $\textbf{31.46}_{\pm3.25}$ | $\textbf{61.00}_{\pm10.10}$ | $\textbf{83.42}_{\pm1.01}$ |
| Neo-GNN | $86.07_{\pm1.96}$ | $83.54_{\pm3.92}$ | $44.04_{\pm1.89}$ | $\textbf{83.14}_{\pm0.73}$ | $63.22_{\pm4.32}$ | $21.98_{\pm4.62}$ | $42.81_{\pm4.13}$ | $73.76_{\pm1.94}$ |
| ELPH | $\textbf{87.60}_{\pm1.49}$ | $\textbf{88.49}_{\pm2.14}$ | $\textbf{46.91}_{\pm2.21}$ | $82.74_{\pm1.19}$ | $\textbf{64.45}_{\pm3.91}$ | $\textbf{26.61}_{\pm1.73}$ | $\textbf{61.07}_{\pm3.06}$ | $75.25_{\pm1.44}$ |
| NCNC | $86.16_{\pm1.77}$ | $83.18_{\pm3.17}$ | $46.85_{\pm3.18}$ | $82.00_{\pm0.97}$ | $60.49_{\pm5.09}$ | $23.28_{\pm1.55}$ | $52.45_{\pm8.77}$ | $\textbf{83.94}_{\pm1.57}$ |
| MPLP | $\textbf{92.05}_{\pm1.20}$ | $\textbf{89.47}_{\pm1.98}$ | $\textbf{52.55}_{\pm2.90}$ | $\textbf{85.36}_{\pm0.68}$ | $\textbf{74.29}_{\pm2.78}$ | $\textbf{32.25}_{\pm1.43}$ | $60.83_{\pm1.97}$ | $\textbf{87.11}_{\pm0.83}$ |

## 4.3 TRIANGLE ESTIMATIONS

Constructing the structural feature with DE can provably enhance the expressiveness of the link prediction model (Li et al., 2020; Zhang et al., 2021). However, there are still prominent cases where labelling trick also fails to capture. Since labelling trick only considers the relationship between the neighbors and the target node pair, it can sometimes miss the subtleties of intra-neighbor relationships. For example, the nodes of $DE(1, 1)$ in Figure 3 exhibit different local structures. Nevertheless, labelling trick like DE tends to treat them equally, which makes the model overlook the triangle substructure shown in the neighborhood. Chen et al. (2020) discusses the challenge of counting such a substructure with a pure message-passing framework. We next give an implementation of message-passing to approximate triangle counts linked to a target node pair—equivalent in complexity to conventional MPNNs.

For a triangle to form, two nodes must connect with each other and the target node. Key to our methodology is recognizing the obligatory presence of length-1 and length-2 walks to the target node. Thus, according to Theorem 3, our estimation can formalize as:

$$\#_{u}(\triangle) = \frac{1}{2}\mathbb{E}\Big(\tilde{\boldsymbol{h}}_{u}^{(1)} \cdot \tilde{\boldsymbol{h}}_{u}^{(2)}\Big). \tag{7}$$

Augmenting the node label counts with triangle estimates gives rise to a more expressive structural feature set of MPLP.

**Feature integration for link prediction.** Having procured the structural features, we proceed to formulate the encompassing link representation for a target node pair $(u, v)$ as:

$$\boldsymbol{h}_{(u,v)} = (\text{GNN}(u) \odot \text{GNN}(v)) || [\#(1, 1), \dots, \#(r, r), \#_{u}(\triangle), \#_{v}(\triangle)], \tag{8}$$

which can be fed into a classifier for a link prediction between nodes $(u, v)$.

## 5 EXPERIMENTS

**Datasets, baselines and experimental setup** We evaluate our approach on a diverse set of 8 non-attributed and 5 attributed graph benchmarks. In the absence of predefined train/test splits, links are partitioned into train, validation, and test splits following a 70-10-20 percentage distribution. Our comparison spans three categories of link prediction models: (1) heuristic-based methods encompassing CN, AA, and RA; (2) node-level models like GCN and SAGE; and (3) link-level models, including SEAL, Neo-GNN (Yun et al., 2021), ELPH (Chamberlain et al., 2022), and NCNC (Wang et al., 2023). Each experiment is conducted 10 times, with the average score and standard deviations reported using the Hits@50 metric, a well-accepted standard for the link prediction task (Hu et al., 2021). We limit the number of hops $r = 2$, which results in a good balance of performance and efficiency. A comprehensive description of the experimental setup is available in Appendix B.

**Results** Performance metrics are presented in Table 1 and Table 2. MPLP outperforms other models on 12 of the 13 benchmarks. In the context of non-attributed graphs, MPLP takes the lead on 7 out of the 8 datasets, followed by SEAL and ELPH. For attributed graphs, MPLP reigns supreme on all 5 datasets. Notably, MPLP consistently demonstrates superior results across a wide range of graph domains, with a performance advantage ranging from $2\%$ to $10\%$ in Hits@50 over the closest competitors. More ablation study can be found in Appendix D.

Table 2: Link prediction results on attributed benchmarks evaluated by Hits@50. The format is average score ± standard deviation. The top three models are colored by **First**, **Second**, **Third**.

Figure 4: Evaluation of model size and inference time on Collab. The inference time encompasses the entire cycle within a single epoch.

|  | CS | Physics | Computers | Photo | Collab |
|---|---|---|---|---|---|
| **CN** | $51.04_{\pm15.56}$ | $61.46_{\pm6.12}$ | $21.95_{\pm2.00}$ | $29.33_{\pm2.74}$ | $61.37_{\pm0.00}$ |
| **AA** | $68.26_{\pm1.28}$ | $70.98_{\pm1.96}$ | $26.96_{\pm2.08}$ | $37.35_{\pm2.65}$ | $64.35_{\pm0.00}$ |
| **RA** | $68.25_{\pm1.29}$ | $72.29_{\pm1.69}$ | $28.05_{\pm1.59}$ | $40.77_{\pm3.41}$ | $64.00_{\pm0.00}$ |
| **GCN** | $66.00_{\pm2.90}$ | $73.71_{\pm2.28}$ | $22.95_{\pm10.58}$ | $28.14_{\pm7.81}$ | $35.53_{\pm2.39}$ |
| **SAGE** | $57.79_{\pm18.23}$ | $74.10_{\pm2.51}$ | $33.79_{\pm3.11}$ | $46.01_{\pm1.83}$ | $36.82_{\pm7.41}$ |
| **SEAL** | $68.50_{\pm0.76}$ | $74.27_{\pm2.58}$ | $30.43_{\pm2.07}$ | $46.08_{\pm3.27}$ | $64.74_{\pm0.43}$ |
| **Neo-GNN** | $71.13_{\pm1.69}$ | $72.28_{\pm2.33}$ | $22.76_{\pm3.07}$ | $44.83_{\pm3.23}$ | $57.52_{\pm0.37}$ |
| **ELPH** | $72.26_{\pm2.58}$ | $65.80_{\pm2.26}$ | $29.01_{\pm2.66}$ | $43.51_{\pm2.37}$ | $65.94_{\pm0.58}$ |
| **NCNC** | $74.65_{\pm1.23}$ | $75.96_{\pm1.73}$ | $36.48_{\pm4.16}$ | $47.98_{\pm2.36}$ | $66.61_{\pm0.71}$ |
| **MPLP** | $76.40_{\pm1.44}$ | $76.06_{\pm2.31}$ | $40.51_{\pm2.91}$ | $56.50_{\pm2.82}$ | $67.05_{\pm0.51}$ |

Figure 5: MSE of estimation for $\#(1,1)$, $\#(1,2)$ and $\#(1,0)$ on Collab. Lower values are better.

**Model size and inference time**   A separate assessment focuses on the trade-off between model size and inference time using the Collab dataset, with findings presented in Figure 4. Observing the prominent role of graph structure in link prediction performance on Collab, we introduce a streamlined version of our model, termed MPLP(no feat). This variant solely capitalizes on structural features, resulting in a compact model with merely 260 parameters. Nevertheless, its efficacy rivals that of models which are orders of magnitude larger. Furthermore, MPLP's inference time for a single epoch ranks among the quickest in state-of-the-art approaches, underscoring its efficiency both in terms of time and memory footprint. More details can be found in Appendix B.3.

**Estimation accuracy**   We investigate the precision of MPLP in estimating $\#(p,q)$, which denotes the count of node labels, using the Collab dataset. The outcomes of this examination are illustrated in Figure 5. Although ELPH possesses the capability to approximate these counts utilizing techniques like MinHash and Hyperloglog, our method exhibits superior accuracy. Moreover, ELPH runs out of memory when the dimension is larger than 3000. Remarkably, deploying a one-hot encoding strategy for the hubs further bolsters the accuracy of MPLP, concurrently diminishing the variance introduced by inherent graph structures. An exhaustive analysis, including time efficiency considerations, is provided in Appendix D.1.

# 6   CONCLUSION

In this work, we delved into the potential of message-passing GNNs to encapsulate joint structural features of graphs. Stemming from this investigation, we introduced a novel link prediction paradigm that consistently outperforms state-of-the-art baselines across a varied suite of graph benchmarks. The inherent capability to adeptly capture structures enhances the expressivity of GNNs, all while maintaining their computational efficiency. Our findings hint at a promising avenue for elevating the expressiveness of GNNs through probabilistic approaches.

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

# A  RELATED WORK

**Link prediction**    Link prediction, inherent to graph data analysis, has witnessed a paradigm shift from its conventional heuristic-based methods to the contemporary, more sophisticated GNNs approaches. Initial explorations in this domain primarily revolve around heuristic methods such as CN, AA, RA, alongside seminal heuristics like the Katz Index (Katz, 1953), Jaccard Index (Salton & McGill, 1986), Page Rank (Brin & Page, 1998), and Preferential Attachment (Barabási & Albert, 1999). However, the emergence of graphs associated with node attributes has shifted the research landscape towards GNN-based methods. Specifically, these GNN-centric techniques bifurcate into node-level and link-level paradigms. Pioneers like Kipf & Welling introduce the Graph Auto-Encoder (GAE) to ascertain node pair similarity through GNN-generated node representation. On the other hand, link-level models, represented by SEAL (Zhang & Chen, 2018), opt for subgraph extractions centered on node pairs, even though this can present scalability challenges.

**Amplifying GNN Expressiveness with Randomness**    The expressiveness of GNNs, particularly those of the MPNNs, has been the subject of rigorous exploration (Xu et al., 2018). A known limitation of MPNNs, their equivalence to the 1-Weisfeiler-Lehman test, often results in indistinguishable representation for non-isomorphic graphs. A suite of contributions has surfaced to boost GNN expressiveness, of which (Morris et al., 2021; Maron et al., 2020; Zhang & Li, 2021; Frasca et al., 2022) stand out. An elegant, yet effective paradigm involves symmetry-breaking through stochasticity injection (Sato et al., 2021; Abboud et al., 2021; Papp et al., 2021). Although enhancing expressiveness, such random perturbations can occasionally undermine generalizability. Diverging from these approaches, our methodology exploits probabilistic orthogonality within random vectors, culminating in a robust structural feature estimator that introduces minimal estimator variance.

**Link-Level Link Prediction**    While node-level models like GAE offer enviable efficiency, they occasionally fall short in performance when compared with rudimentary heuristics (Chamberlain et al., 2022). Efforts to build scalable link-level alternatives have culminated in innovative methods such as Neo-GNN (Yun et al., 2021), which distills structural features from adjacency matrices for link prediction. Elsewhere, ELPH (Chamberlain et al., 2022) harnesses hashing mechanisms for structural feature representation, while NCNC (Wang et al., 2023) adeptly aggregates common neighbors' node representation. Notably, DotHash (Nunes et al., 2023), which profoundly influenced our approach, employs quasi-orthogonal random vectors for set similarity computations, applying these in link prediction tasks.

Distinctively, our proposition builds upon, yet diversifies from, the frameworks of ELPH and DotHash. While resonating with ELPH's architectural spirit, we utilize a streamlined, efficacious hashing technique over MinHash for set similarity computations. Moreover, we resolve ELPH's limitations through strategic implementations like shortcut removal and norm rescaling. When paralleled with DotHash, our approach magnifies its potential, integrating it with GNNs for link predictions and extrapolating its applicability to multi-hop scenarios. It also judiciously optimizes variance induced by the structural feature estimator in sync with graph data. We further explore the potential of achieving higher expressiveness with linear computational complexity by estimating the substructure counting (Chen et al., 2020).

# B  EXPERIMENTAL DETAILS

## B.1  BENCHMARK DATASETS

The statistics of each benchmark dataset are shown in Table 3. The benchmarks without attributes are:

- **USAir** (Batagelj & Mrvar, 2006): a graph of US airlines;
- **NS** (Newman, 2006): a collaboration network of network science researchers;
- **PB** (Ackland & others, 2005): a graph of links between web pages on US political topics;
- **Yeast** (Von Mering et al., 2002): a protein-protein interaction network in yeast;
- **C.ele** (Watts & Strogatz, 1998): the neural network of Caenorhabditis elegans;

Table 3: Statistics of benchmark datasets.

| Dataset | #Nodes | #Edges | Avg. node deg. | Std. node deg. | Max. node deg. | Density | Attr. Dimension |
|---|---|---|---|---|---|---|---|
| C.ele | 297 | 4296 | 14.46 | 12.97 | 134 | 9.7734% | - |
| Yeast | 2375 | 23386 | 9.85 | 15.50 | 118 | 0.8295% | - |
| Power | 4941 | 13188 | 2.67 | 1.79 | 19 | 0.1081% | - |
| Router | 5022 | 12516 | 2.49 | 5.29 | 106 | 0.0993% | - |
| USAir | 332 | 4252 | 12.81 | 20.13 | 139 | 7.7385% | - |
| E.coli | 1805 | 29320 | 16.24 | 48.38 | 1030 | 1.8009% | - |
| NS | 1589 | 5484 | 3.45 | 3.47 | 34 | 0.4347% | - |
| PB | 1222 | 33428 | 27.36 | 38.42 | 351 | 4.4808% | - |
| CS | 18333 | 163788 | 8.93 | 9.11 | 136 | 0.0975% | 6805 |
| Physics | 34493 | 495924 | 14.38 | 15.57 | 382 | 0.0834% | 8415 |
| Computers | 13752 | 491722 | 35.76 | 70.31 | 2992 | 0.5201% | 767 |
| Photo | 7650 | 238162 | 31.13 | 47.28 | 1434 | 0.8140% | 745 |
| Collab | 235868 | 2358104 | 10.00 | 18.98 | 671 | 0.0085% | 128 |

- **Power** (Watts & Strogatz, 1998): the network of the western US's electric grid;
- **Router** (Spring et al., 2002): the Internet connection at the router-level;
- **E.coli** (Zhang et al., 2018): the reaction network of metabolites in Escherichia coli.

4 out of 5 benchmarks with node attributes come from (Shchur et al., 2019), while Collab is from Open Graph Benchmark (Hu et al., 2021):

- **CS**: co-authorship graphs in the field of computer science, where nodes represent authors, edges represent that two authors collaborated on a paper, and node features indicate the keywords for each author's papers;
- **Physics**: co-authorship graphs in the field of physics with the same node/edge/feature definition as of **CS**;
- **Computers**: a segment of the Amazon co-purchase graph for computer-related equipment, where nodes represent goods, edges represent that two goods are frequently purchased together together, and node features represent the product reviews;
- **Physics**: a segment of the Amazon co-purchase graph for photo-related equipment with the same node/edge/feature definition as of **Computers**;
- **Collab**: a large-scale collaboration network, showcasing a wide array of interdisciplinary partnerships.

Since Collab has a fixed split, no train test split is needed for it. For the other benchmarks, we randomly split the edges into 70-10-20 as train, validation, and test sets. The validation and test sets are not observed in the graph during the entire cycle of training and testing. They are only used for evaluation purposes. For Collab, it is allowed to use the validation set in the graph when evaluating on the test set.

We run the experiments 10 times on each dataset with different splits. For each run, we cache the split edges and evaluate every model on the same split to ensure a fair comparison. The average score and standard deviation are reported for Hits@50.

## B.2    MORE DETAILS IN BASELINE METHODS

In our experiments, we explore advanced variants of the baseline models **ELPH** and **NCNC**. Specifically, for **ELPH**, Chamberlain et al. (2022) propose BUDDY, an enhanced link prediction method that preprocesses node representations for efficiency. **NCNC** (Wang et al., 2023) builds upon its predecessor, NCN, by first estimating the complete graph structure and then performing inference.

We incorporate these latest and most accurate versions of both models to establish robust baselines in our study.

### B.3 EVALUATION DETAILS: INFERENCE TIME

In Figure 4, we assess the inference time across different models on the Collab dataset for a single epoch of test links. Specifically, we clock the wall time taken by models to score the complete test set. This encompasses preprocessing, message-passing, and the actual prediction. For the SEAL model, we employ a dynamic subgraph generator during the preprocessing phase, which dynamically computes the subgraph. Meanwhile, for both ELPH and our proposed method, MPLP, we initially propagate the node features and signatures just once at the onset of inference. These are then cached for subsequent scoring sessions.

### B.4 SOFTWARE AND HARDWARE DETAILS

We implement MPLP in Pytorch Geometric framework (Fey & Lenssen, 2019). We run our experiments on a Linux system equipped with an NVIDIA V100 GPU with 32GB of memory.

### B.5 TIME COMPLEXITY

The efficiency of MPLP stands out when it comes to link prediction inference. Let's denote $t$ as the number of target links, $d$ as the maximum node degree, $r$ as the number of hops to compute, and $F$ as the dimension count of node signatures.

For preprocessing node signatures, MPLP involves two primary steps:

1. Initially, the algorithm computes all-pairs unweighted shortest paths across the input graph to acquire the shortest-path neighborhood $\mathcal{N}_v^s$ for each node. This can be achieved using a BFS approach for each node, with a time complexity of $O(|V||E|)$.
2. Following this, MPLP propagates the QO vectors through the shortest-path neighborhood, which has a complexity of $O(td^r F)$, and then caches these vectors in memory.

During online scoring, MPLP performs the inner product operation with a complexity of $O(tF)$, enabling the extraction of structural feature estimations.

However, during training, the graph's structure might vary depending on the batch of target links due to the shortcut removal operation. As such, MPLP proceeds in three primary steps:

1. Firstly, the algorithm extracts the $r$-hop induced subgraph corresponding to these $t$ target links. In essence, we deploy a BFS starting at each node of the target links to determine their receptive fields. This process, conceptually similar to message-passing but in a reversed message flow, has a time complexity of $O(tdr)$. Note that, different from SEAL, we extract one $r$-hop subgraph induced from a batch of target links.
2. To identify the shortest-path neighborhood $\mathcal{N}_v^s$, we simply apply sparse-sparse matrix multiplications of the adjacency matrix to get the $s$-power adjacency matrix, where $s = 1, 2, \ldots, r$. Due to the sparsity, this takes $O(|V|d^r)$.
3. Finally, the algorithm engages in message-passing to propagate the QO vectors along the shortest-path neighborhoods, with a complexity of $O(td^r F)$, followed by performing the inner product at $O(tF)$.

Summing up, the overall time complexity for the training phase stands at $O(tdr + |V|d^r + td^r F)$.

### B.6 HYPERPARAMETERS

We determine the optimal hyperparameters for our model through systematic exploration. The setting with the best performance on the validation set is selected. The chosen hyperparameters are as follows:

- Number of Hops ($r$): We set the maximum number of hops to $r = 2$. Empirical evaluation suggests this provides an optimal trade-off between accuracy and computational efficiency.

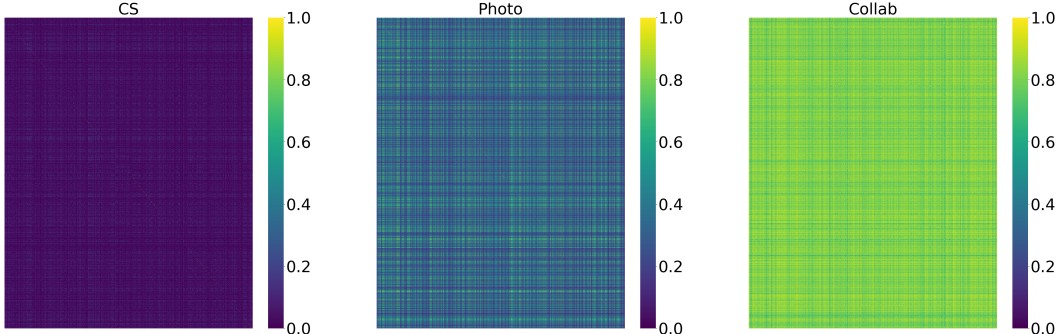

Figure 6: Heatmap illustrating the inner product of node attributes across CS, Photo, and Collab datasets.

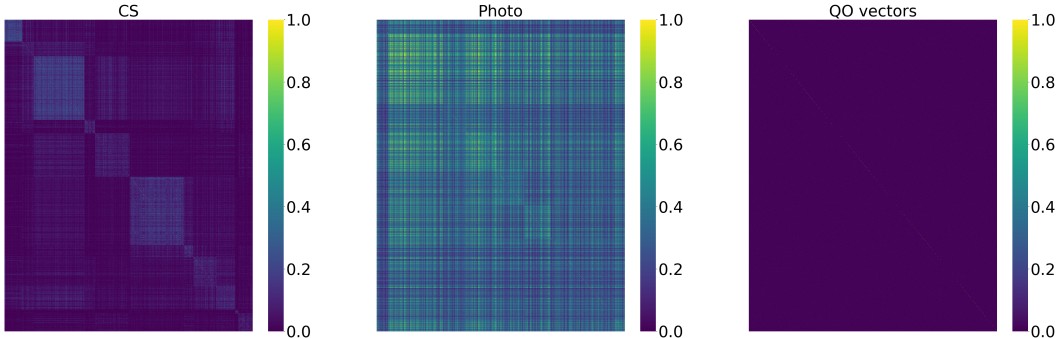

Figure 7: Heatmap illustrating the inner product of node attributes, arranged by node labels, across CS and Photo. The rightmost showcases the inner product of QO vectors.

- Node Signature Dimension ($F$): The dimension of node signatures, $F$, is fixed at $1024$, except for Collab with $2048$. This configuration ensures that MPLP is both efficient and accurate across all benchmark datasets.

- The minimum degree of nodes to be considered as hubs ($b$): This parameter indicates the minimum degree of the nodes which are considered as hubs to one-hot encode in the node signatures. We experiment with values in the set $[50, 100, 150]$.

- Batch Size ($B$): We vary the batch size depending on the graph type: For the 8 non-attributed graphs, we explore batch sizes within $[512, 1024]$. For the 5 attributed graphs, we extend our search to $[2048, 4096]$.

More ablation study can be found in Appendix D.4.

## C  EXPLORING BAG-OF-WORDS NODE ATTRIBUTES

In Section 3, we delved into the capability of GNNs to discern joint structural features, particularly when presented with Quasi-Orthogonal (QO) vectors. Notably, many graph benchmarks utilize text data to construct node attributes, representing them as Bag-Of-Words (BOW). BOW is a method that counts word occurrences, assigning these counts as dimensional values. With a large dictionary, these BOW node attribute vectors often lean towards QO due to the sparse nature of word representations. Consequently, many node attributes in graph benchmarks inherently possess the QO trait. Acknowledging GNNs' proficiency with QO vector input, we propose the question: *Is it the **QO** property or the **information** embedded within these attributes that significantly impacts link prediction in benchmarks?* This section is an empirical exploration of this inquiry.

Table 4: Performance comparison of GNNs using node attributes versus random vectors (Hits@50). For simplicity, all GNNs are configured with two layers.

|  | CS | Physics | Computers | Photo | Collab |
|---|---|---|---|---|---|
| **GCN** | $66.00_{\pm 2.90}$ | $73.71_{\pm 2.28}$ | $22.95_{\pm 10.58}$ | $28.14_{\pm 7.81}$ | $35.53_{\pm 2.39}$ |
| **GCN(random feat)** | $51.67_{\pm 2.70}$ | $69.55_{\pm 2.45}$ | $35.86_{\pm 3.17}$ | $46.84_{\pm 2.53}$ | $17.25_{\pm 1.15}$ |
| **SAGE** | $57.79_{\pm 18.23}$ | $74.10_{\pm 2.51}$ | $1.86_{\pm 2.53}$ | $5.70_{\pm 10.15}$ | $36.82_{\pm 7.41}$ |
| **SAGE(random feat)** | $11.78_{\pm 1.62}$ | $64.71_{\pm 3.65}$ | $29.23_{\pm 3.92}$ | $39.94_{\pm 3.41}$ | $28.87_{\pm 2.36}$ |
| **Random feat** | | | | | |
| **GCN**($F = 1000$) | $3.73_{\pm 1.44}$ | $49.28_{\pm 2.74}$ | $36.92_{\pm 3.36}$ | $48.72_{\pm 3.84}$ | $31.93_{\pm 2.10}$ |
| **GCN**($F = 2000$) | $24.97_{\pm 2.67}$ | $49.13_{\pm 4.64}$ | $40.24_{\pm 3.04}$ | $53.49_{\pm 3.50}$ | $40.16_{\pm 1.70}$ |
| **GCN**($F = 3000$) | $39.51_{\pm 6.47}$ | $53.76_{\pm 3.85}$ | $42.33_{\pm 3.82}$ | $56.27_{\pm 3.47}$ | $47.22_{\pm 1.60}$ |
| **GCN**($F = 4000$) | $43.23_{\pm 3.37}$ | $61.86_{\pm 4.10}$ | $42.85_{\pm 3.60}$ | $56.87_{\pm 3.59}$ | $50.40_{\pm 1.28}$ |
| **GCN**($F = 5000$) | $48.25_{\pm 3.28}$ | $63.19_{\pm 4.31}$ | $44.52_{\pm 2.78}$ | $58.13_{\pm 3.79}$ | $52.13_{\pm 1.02}$ |
| **GCN**($F = 6000$) | $51.44_{\pm 1.50}$ | $65.10_{\pm 4.11}$ | $44.90_{\pm 2.74}$ | $58.10_{\pm 3.35}$ | $53.78_{\pm 0.84}$ |
| **GCN**($F = 7000$) | $52.00_{\pm 1.74}$ | $66.76_{\pm 3.32}$ | $45.11_{\pm 3.69}$ | $57.41_{\pm 2.62}$ | $55.04_{\pm 1.06}$ |
| **GCN**($F = 8000$) | $54.21_{\pm 3.47}$ | $69.27_{\pm 2.94}$ | $44.47_{\pm 4.11}$ | $58.67_{\pm 3.90}$ | $55.36_{\pm 1.15}$ |
| **GCN**($F = 9000$) | $53.16_{\pm 2.80}$ | $70.79_{\pm 2.83}$ | $45.03_{\pm 3.13}$ | $57.15_{\pm 3.87}$ | OOM |
| **GCN**($F = 10000$) | $55.91_{\pm 2.63}$ | $71.88_{\pm 3.29}$ | $45.26_{\pm 1.94}$ | $58.12_{\pm 2.54}$ | OOM |

## C.1 NODE ATTRIBUTE ORTHOGONALITY

Our inquiry begins with the assessment of node attribute orthogonality across three attributed graphs: CS, Photo, and Collab. CS possesses extensive BOW vocabulary, resulting in node attributes spanning over 8000 dimensions. Contrarily, Photo has a comparatively minimal dictionary, encompassing just 745 dimensions. Collab, deriving node attributes from word embeddings, limits to 128 dimensions.

For our analysis, we sample 10000 nodes (7650 for Photo) and compute the inner product of their attributes. The results are visualized in Figure 6. Our findings confirm that with a larger BOW dimension, CS node attributes closely follow QO. However, this orthogonality isn't as pronounced in Photo and Collab—especially Collab, where word embeddings replace BOW. Given that increased node signature dimensions can mitigate estimation variance (as elaborated in Theorem 2), one could posit GNNs might offer enhanced performance on CS, due to its extensive BOW dimensions. Empirical evidence from Table 2 supports this claim.

Further, in Figure 7, we showcase the inner product of node attributes in CS and Photo, but this time, nodes are sequenced by class labels. This order reveals that nodes sharing labels tend to have diminished orthogonality compared to random pairs—a potential variance amplifier in structural feature estimation using node attributes.

## C.2 ROLE OF NODE ATTRIBUTE INFORMATION

To discern the role of embedded information within node attributes, we replace the original attributes in CS, Photo, and Collab with random vectors—denoted as *random feat*. These vectors maintain the original attribute dimensions, though each dimension gets randomly assigned values from $\{-1, 1\}$. The subsequent findings are summarized in Table 4. Intriguingly, even with this "noise" as input, performance remains largely unaltered. CS attributes appear to convey valuable insights for link predictions, but the same isn't evident for the other datasets. In fact, introducing random vectors to Computers and Photo resulted in enhanced outcomes, perhaps due to their original attribute's insufficient orthogonality hampering effective structural feature capture. Collab shows a performance drop with random vectors, implying that the original word embedding can contribute more to the link prediction than structural feature estimation with merely 128 QO vectors.

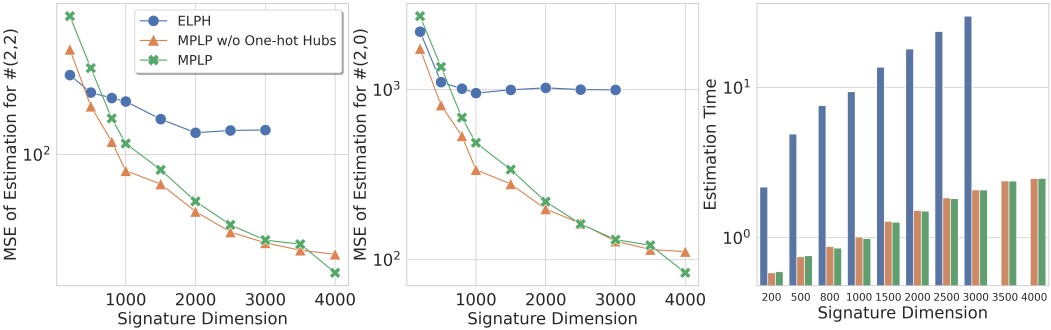

Figure 8: MSE of estimation for #(2, 2), #(2, 0) and estimation time on Collab. Lower values are better.

### C.3 EXPANDING QO VECTOR DIMENSIONS

Lastly, we substitute node attributes with QO vectors of varied dimensions, utilizing GCN as the encoder. The outcomes of this experiment are cataloged in Table 4. What's striking is that GCNs, when furnished with lengthier random vectors, often amplify link prediction results across datasets, with the exception of CS. On Computers and Photo, a GCN even rivals our proposed model (Figure 2), potentially attributed to the enlarged vector dimensions. This suggests that when computational resources permit, expanding our main experiment's node signature dimensions (currently set at $1024$) could elevate our model's performance. On Collab. the performance increases significantly compared to the experiments which are input with 128-dimensional vectors, indicating that the structural features are more critical for Collab than the word embedding.

## D   ADDITIONAL EXPERIMENTS

### D.1   NODE LABEL ESTIMATION ACCURACY AND TIME

In Figure 5, we assess the accuracy of node label count estimation. For ELPH, the node signature dimension corresponds to the number of MinHash permutations. We employ a default hyperparameter setting for Hyperloglog, with $p = 8$, a configuration that has demonstrated its adequacy in (Chamberlain et al., 2022). For time efficiency evaluation, we initially propagate and cache node signatures, followed by performing the estimation.

Furthermore, we evaluate the node label count estimation for #(2, 2) and #(2, 0). The outcomes are detailed in Figure 8. While MPLP consistently surpasses ELPH in estimation accuracy, the gains achieved via one-hot hubs diminish for #(2, 2) and #(2, 0) relative to node counts at a shortest-path distance of $1$. This diminishing performance gain can be attributed to our selection criteria for one-hot encoding, which prioritizes nodes that function as hubs within a one-hop radius. However, one-hop hubs don't necessarily serve as two-hop hubs. While we haven't identified a performance drop for these two-hop node label counts, an intriguing avenue for future research would be to refine variance reduction strategies for both one-hop and two-hop estimations simultaneously.

Regarding the efficiency of estimation, MPLP consistently demonstrates superior computational efficiency in contrast to ELPH. When we increase the node signature dimension to minimize estimation variance, ELPH's time complexity grows exponentially and becomes impractical. In contrast, MPLP displays a sublinear surge in estimation duration.

It's also worth noting that ELPH exhausts available memory when the node signature dimension surpasses 3000. This constraint arises as ELPH, while estimating structural features, has to cache node signatures for both MinHash and Hyperloglog. Conversely, MPLP maintains efficiency by caching only one type of node signatures.

Table 5: Ablation study on non-attributed benchmarks evaluated by Hits@50. The format is average score ± standard deviation. The top three models are colored by **First**, **Second**, **Third**.

| | USAir | NS | PB | Yeast | C.ele | Power | Router | E.coli |
|---|---|---|---|---|---|---|---|---|
| **w/o Shortcut removal** | $80.94_{\pm 3.49}$ | $85.47_{\pm 2.60}$ | $49.51_{\pm 3.57}$ | $82.62_{\pm 0.99}$ | $57.51_{\pm 2.09}$ | $19.99_{\pm 2.54}$ | $36.67_{\pm 10.03}$ | $76.94_{\pm 1.54}$ |
| **w/o One-hot hubs** | $84.04_{\pm 4.53}$ | $89.45_{\pm 2.60}$ | $51.49_{\pm 2.63}$ | $85.11_{\pm 0.62}$ | $66.85_{\pm 3.08}$ | $29.54_{\pm 1.79}$ | $50.81_{\pm 3.74}$ | $79.07_{\pm 2.47}$ |
| **w/o Norm rescaling** | $85.04_{\pm 2.64}$ | $89.34_{\pm 2.79}$ | $52.50_{\pm 2.90}$ | $83.01_{\pm 1.03}$ | $66.81_{\pm 4.11}$ | $29.00_{\pm 2.30}$ | $50.43_{\pm 3.59}$ | $79.36_{\pm 2.18}$ |
| **MPLP** | $85.19_{\pm 4.59}$ | $89.58_{\pm 2.60}$ | $52.84_{\pm 3.39}$ | $85.11_{\pm 0.62}$ | $67.97_{\pm 2.96}$ | $29.54_{\pm 1.79}$ | $51.04_{\pm 4.03}$ | $79.35_{\pm 2.35}$ |

Table 6: Ablation study on attributed benchmarks evaluated by Hits@50. The format is average score ± standard deviation. The top three models are colored by **First**, **Second**, **Third**.

| | CS | Physics | Computers | Photo | Collab |
|---|---|---|---|---|---|
| **w/o Shortcut removal** | $41.63_{\pm 7.27}$ | $62.58_{\pm 2.40}$ | $32.74_{\pm 3.03}$ | $52.09_{\pm 2.52}$ | $60.45_{\pm 1.44}$ |
| **w/o One-hot hubs** | $65.49_{\pm 4.28}$ | $71.58_{\pm 2.28}$ | $36.09_{\pm 4.08}$ | $55.63_{\pm 2.48}$ | $65.07_{\pm 0.47}$ |
| **w/o Norm rescaling** | $65.20_{\pm 2.92}$ | $67.73_{\pm 2.54}$ | $35.83_{\pm 3.24}$ | $52.59_{\pm 3.57}$ | $63.99_{\pm 0.59}$ |
| **MPLP** | $65.70_{\pm 3.86}$ | $71.03_{\pm 3.55}$ | $37.56_{\pm 3.57}$ | $55.63_{\pm 2.48}$ | $66.07_{\pm 0.47}$ |

## D.2 MODEL ENHANCEMENT ABLATION

We investigate the individual performance contributions of three primary components in MPLP: Shortcut removal, One-hot hubs, and Norm rescaling. To ensure a fair comparison, we maintain consistent hyperparameters across benchmark datasets, modifying only the specific component under evaluation. Moreover, node attributes are excluded from the model's input for this analysis. The outcomes of this investigation are detailed in Table 5 and Table 6.

Among the three components, Shortcut removal emerges as the most pivotal for MPLP. This highlights the essential role of ensuring the structural distribution of positive links is aligned between the training and testing datasets (Dong et al., 2022).

Regarding One-hot hubs, while they exhibited strong results in the estimation accuracy evaluations presented in Figure 5 and Figure 8, their impact on the overall performance is relatively subdued. We hypothesize that, in the context of these sparse benchmark graphs, the estimation variance may not be sufficiently influential on the model's outcomes.

Finally, Norm rescaling stands out as a significant enhancement in MPLP. This is particularly evident in its positive impact on datasets like Yeast, Physics, Photo, and Collab.

## D.3 STRUCTURAL FEATURES ABLATION

We further examine the contribution of various structural features to the link prediction task. These features include: $\#(1, 1)$, $\#(1, 2)$, $\#(1, 0)$, $\#(2, 2)$, $\#(2, 0)$, and $\#(\triangle)$. To ensure fair comparison, we utilize only the structural features for link representation, excluding the node representations derived from $\text{GNN}(\cdot)$. Given the combinatorial nature of these features, they are grouped into four categories:

- $\#(1, 1)$;
- $\#(1, 2)$, $\#(1, 0)$;
- $\#(2, 2)$, $\#(2, 0)$;
- $\#(\triangle)$.

The configuration of these structural features and their corresponding results are detailed in Table 7 and Table 8.

Our analysis reveals that distinct benchmark datasets have varied preferences for structural features, reflecting their unique underlying distributions. For example, datasets PB and Power exhibit superior performance with 2-hop structural features, whereas others predominantly favor 1-hop features. Although $\#(1, 1)$, which counts Common Neighbors, is often considered pivotal for link prediction, the two other 1-hop structural features, $\#(1, 2)$ and $\#(1, 0)$, demonstrate a more pronounced impact

Table 7: The mapping between the configuration number and the used structural features in MPLP.

| Configurations | #(1,1) | #(1,2) | #(1,0) | #(2,2) | #(2,0) | #(△) |
|---|---|---|---|---|---|---|
| **(1)** | ✓ | - | - | - | - | - |
| **(2)** | - | ✓ | ✓ | - | - | - |
| **(3)** | - | - | - | ✓ | ✓ | - |
| **(4)** | - | - | - | - | - | ✓ |
| **(5)** | ✓ | ✓ | ✓ | - | - | - |
| **(6)** | ✓ | - | - | ✓ | ✓ | - |
| **(7)** | ✓ | - | - | - | - | ✓ |
| **(8)** | - | ✓ | ✓ | ✓ | ✓ | - |
| **(9)** | - | ✓ | ✓ | - | - | ✓ |
| **(10)** | - | - | - | ✓ | ✓ | ✓ |
| **(11)** | ✓ | ✓ | ✓ | ✓ | ✓ | - |
| **(12)** | ✓ | ✓ | ✓ | - | - | ✓ |
| **(13)** | ✓ | - | - | ✓ | ✓ | ✓ |
| **(14)** | - | ✓ | ✓ | ✓ | ✓ | ✓ |
| **(15)** | ✓ | ✓ | ✓ | ✓ | ✓ | ✓ |

Table 8: Ablation analysis highlighting the impact of various structural features on link prediction. Refer to Table 7 for detailed configurations of the structural features used.

| Configurations | USAir | NS | PB | Yeast | C.ele | Power | Router | E.coli |
|---|---|---|---|---|---|---|---|---|
| **(1)** | $76.64_{\pm26.74}$ | $75.26_{\pm2.79}$ | $37.48_{\pm13.30}$ | $58.70_{\pm30.50}$ | $46.22_{\pm24.84}$ | $14.40_{\pm1.40}$ | $17.29_{\pm3.96}$ | $60.10_{\pm30.80}$ |
| **(2)** | $82.54_{\pm4.61}$ | $84.76_{\pm3.63}$ | $41.84_{\pm15.51}$ | $80.56_{\pm0.65}$ | $56.22_{\pm20.39}$ | $21.38_{\pm1.46}$ | $48.97_{\pm3.34}$ | $67.78_{\pm23.83}$ |
| **(3)** | $67.76_{\pm23.65}$ | $70.05_{\pm2.35}$ | $44.81_{\pm2.63}$ | $67.02_{\pm2.53}$ | $36.53_{\pm19.68}$ | $25.24_{\pm4.07}$ | $21.32_{\pm2.66}$ | $56.59_{\pm1.78}$ |
| **(4)** | $37.18_{\pm37.57}$ | $25.13_{\pm1.99}$ | $12.35_{\pm10.75}$ | $7.42_{\pm10.80}$ | $30.75_{\pm18.69}$ | $5.47_{\pm1.13}$ | $30.47_{\pm3.10}$ | $34.90_{\pm36.63}$ |
| **(5)** | $86.24_{\pm2.70}$ | $84.91_{\pm2.80}$ | $48.35_{\pm3.76}$ | $84.42_{\pm0.56}$ | $66.69_{\pm3.60}$ | $22.25_{\pm1.39}$ | $49.68_{\pm3.79}$ | $80.94_{\pm1.62}$ |
| **(6)** | $77.41_{\pm5.27}$ | $80.00_{\pm2.39}$ | $46.05_{\pm2.76}$ | $74.70_{\pm1.45}$ | $46.88_{\pm5.79}$ | $27.74_{\pm3.23}$ | $22.37_{\pm2.06}$ | $71.41_{\pm2.47}$ |
| **(7)** | $71.11_{\pm25.51}$ | $76.72_{\pm2.37}$ | $43.57_{\pm3.70}$ | $73.08_{\pm1.23}$ | $54.99_{\pm20.14}$ | $14.50_{\pm1.64}$ | $31.26_{\pm2.87}$ | $80.22_{\pm2.09}$ |
| **(8)** | $80.16_{\pm4.82}$ | $88.67_{\pm2.72}$ | $52.16_{\pm2.25}$ | $82.52_{\pm0.85}$ | $63.82_{\pm4.02}$ | $28.41_{\pm2.00}$ | $50.97_{\pm3.57}$ | $77.26_{\pm1.31}$ |
| **(9)** | $75.13_{\pm26.51}$ | $87.28_{\pm3.33}$ | $48.10_{\pm3.43}$ | $80.84_{\pm0.97}$ | $60.63_{\pm4.54}$ | $23.85_{\pm1.37}$ | $49.78_{\pm3.56}$ | $76.13_{\pm1.81}$ |
| **(10)** | $76.82_{\pm4.28}$ | $77.04_{\pm3.70}$ | $45.42_{\pm2.77}$ | $67.34_{\pm3.20}$ | $41.66_{\pm13.47}$ | $26.95_{\pm1.47}$ | $28.31_{\pm2.76}$ | $70.14_{\pm0.77}$ |
| **(11)** | $82.82_{\pm5.52}$ | $88.91_{\pm2.90}$ | $52.57_{\pm3.05}$ | $84.61_{\pm0.67}$ | $67.11_{\pm2.52}$ | $28.98_{\pm1.73}$ | $50.63_{\pm3.72}$ | $80.16_{\pm2.20}$ |
| **(12)** | $87.29_{\pm1.08}$ | $88.08_{\pm2.59}$ | $48.86_{\pm3.42}$ | $84.59_{\pm0.69}$ | $66.06_{\pm3.74}$ | $23.79_{\pm1.87}$ | $50.06_{\pm3.66}$ | $79.57_{\pm2.46}$ |
| **(13)** | $78.21_{\pm2.74}$ | $88.08_{\pm3.27}$ | $46.00_{\pm2.31}$ | $74.88_{\pm2.49}$ | $54.64_{\pm4.99}$ | $28.82_{\pm1.29}$ | $26.24_{\pm2.18}$ | $74.67_{\pm3.96}$ |
| **(14)** | $80.75_{\pm5.02}$ | $89.14_{\pm2.38}$ | $51.63_{\pm2.67}$ | $82.68_{\pm0.67}$ | $63.01_{\pm3.21}$ | $29.41_{\pm1.44}$ | $51.08_{\pm4.12}$ | $76.88_{\pm1.86}$ |
| **(15)** | $81.06_{\pm6.62}$ | $89.73_{\pm2.12}$ | $53.49_{\pm2.66}$ | $85.06_{\pm0.69}$ | $66.41_{\pm3.02}$ | $28.86_{\pm2.40}$ | $50.63_{\pm3.79}$ | $78.91_{\pm2.58}$ |

on link prediction outcomes. Meanwhile, while the count of triangles, $\#(\triangle)$, possesses theoretical significance for model expressiveness, it seems less influential for link prediction when assessed in isolation. However, its presence can bolster link prediction performance when combined with other key structural features.

## D.4 PARAMETER SENSITIVITY

We perform an ablation study to assess the hyperparameter sensitivity of MPLP, focusing specifically on two parameters: Batch Size ($B$) and Node Signature Dimension ($F$).

Our heightened attention to $B$ stems from its role during training. Within each batch, MPLP executes the shortcut removal. Ideally, if $B = 1$, only one target link would be removed, thereby

Table 9: Ablation study of Batch Size ($B$) on non-attributed benchmarks evaluated by Hits@50. The format is average score ± standard deviation. The top three models are colored by **First**, **Second**, **Third**.

| | USAir | NS | PB | Yeast | C.ele | Power | Router | E.coli |
|---|---|---|---|---|---|---|---|---|
| **MPLP**($B = 256$) | 90.31±1.32 | 88.98±2.48 | 51.14±2.44 | 84.07±0.69 | 71.59±2.83 | 28.92±1.67 | 56.15±3.80 | 85.12±1.00 |
| **MPLP**($B = 512$) | 90.40±2.47 | 89.40±2.12 | 49.63±2.08 | 84.17±0.60 | 71.72±3.35 | 28.60±1.66 | 53.25±6.57 | 84.72±1.04 |
| **MPLP**($B = 1024$) | 90.49±2.22 | 88.49±2.34 | 50.60±3.40 | 83.67±0.57 | 70.61±4.13 | 28.63±1.60 | 49.75±5.14 | 84.52±1.03 |
| **MPLP**($B = 2048$) | 81.20±2.80 | 61.79±18.55 | 50.34±3.05 | 76.79±6.79 | 31.79±19.88 | 28.45±1.88 | 49.37±3.89 | 84.43±1.28 |
| **MPLP**($B = 4096$) | 81.20±2.80 | 61.79±18.55 | 52.59±2.36 | 58.26±7.20 | 31.54±18.53 | 27.25±3.30 | 50.26±3.89 | 85.15±1.15 |
| **MPLP**($B = 8192$) | 81.20±2.80 | 56.20±21.34 | 51.91±2.08 | 24.47±21.12 | 31.79±19.88 | 17.22±3.17 | 38.67±7.78 | 85.67±0.90 |

Table 10: Ablation study of Batch Size ($B$) on attributed benchmarks evaluated by Hits@50. The format is average score ± standard deviation. The top three models are colored by **First**, **Second**, **Third**.

| | CS | Physics | Computers | Photo |
|---|---|---|---|---|
| **MPLP**($B = 256$) | 74.96±1.87 | 76.06±1.47 | 43.38±2.83 | 57.58±2.92 |
| **MPLP**($B = 512$) | 75.61±2.25 | 75.38±1.79 | 42.95±2.56 | 57.19±2.51 |
| **MPLP**($B = 1024$) | 74.89±2.00 | 74.89±1.97 | 42.69±2.41 | 56.97±3.20 |
| **MPLP**($B = 2048$) | 75.02±2.68 | 75.47±1.68 | 41.39±2.87 | 55.89±3.03 |
| **MPLP**($B = 4096$) | 75.46±1.78 | 74.88±2.57 | 40.65±2.85 | 55.89±2.88 |
| **MPLP**($B = 8192$) | 75.26±1.91 | 74.14±2.17 | 40.00±3.40 | 55.90±2.52 |

preserving the local structures of other links. However, this approach is computationally inefficient. Although shortcut removal can markedly enhance performance and address the distribution shift issue (as elaborated in Appendix D.2), it can also inadvertently modify the graph structure. Thus, striking a balance between computational efficiency and minimal graph structure alteration is essential.

Our findings are delineated in Table 9, Table 10, Table 11, and Table 12. Concerning the batch size, our results indicate that opting for a smaller batch size typically benefits performance. However, if this size is increased past a certain benchmark threshold, there can be a noticeable performance drop. This underscores the importance of pinpointing an optimal batch size for MPLP. Regarding the node signature dimension, our data suggests that utilizing longer QO vectors consistently improves accuracy by reducing variance. This implies that, where resources allow, selecting a more substantial node signature dimension is consistently advantageous.

### D.5 EXPERIMENTAL RESULTS UNDER DIFFERENT METRICS

We extend our model evaluation to include additional metrics such as Hits@20 and Hits@100, with the results detailed in Table 13, Table 14, Table 15, and Table 16. In the Hits@20 metric, MPLP maintains its lead, ranking as the top model in 6 out of 8 non-attributed datasets and excelling in 3 attributed datasets. For Hits@100, MPLP consistently ranks among the top two models across all non-attributed datasets and achieves the best performance in 3 attributed benchmarks, securing second place in the Physics and Collab dataset.

Table 11: Ablation study of Node Signature Dimension ($F$) on non-attributed benchmarks evaluated by Hits@50. The format is average score ± standard deviation. The top three models are colored by **First**, **Second**, **Third**.

| | USAir | NS | PB | Yeast | C.ele | Power | Router | E.coli |
|---|---|---|---|---|---|---|---|---|
| **MPLP**($F = 256$) | 90.64±2.50 | 88.52±3.07 | 50.42±3.86 | 80.63±0.84 | 70.89±4.70 | 25.74±1.59 | 51.84±2.90 | 84.60±0.92 |
| **MPLP**($F = 512$) | 90.49±1.95 | 89.18±2.35 | 51.48±2.63 | 82.41±1.10 | 70.91±4.68 | 27.58±1.80 | 51.98±4.38 | 84.70±1.33 |
| **MPLP**($F = 1024$) | 90.16±1.61 | 89.40±2.12 | 50.60±3.40 | 83.87±1.06 | 70.61±4.13 | 28.88±2.24 | 53.92±2.88 | 84.81±0.85 |
| **MPLP**($F = 2048$) | 90.14±2.24 | 89.36±1.92 | 51.26±1.67 | 84.20±1.02 | 72.24±3.31 | 29.27±1.92 | 54.50±4.52 | 84.58±1.42 |
| **MPLP**($F = 4096$) | 89.95±1.48 | 89.54±2.22 | 51.07±2.87 | 84.89±0.64 | 71.91±3.52 | 29.26±1.51 | 54.71±5.07 | 84.67±0.61 |

Table 12: Ablation study of Node Signature Dimension ($F$) on attributed benchmarks evaluated by Hits@50. The format is average score ± standard deviation. The top three models are colored by **First**, **Second**, **Third**.

| | CS | Physics | Computers | Photo |
|---|---|---|---|---|
| **MPLP($F = 256$)** | $74.90_{\pm 1.88}$ | $73.91_{\pm 1.41}$ | $40.65_{\pm 3.24}$ | $55.13_{\pm 2.98}$ |
| **MPLP($F = 512$)** | $74.67_{\pm 2.63}$ | $74.49_{\pm 2.05}$ | $39.36_{\pm 2.28}$ | $\mathbf{55.93_{\pm 3.31}}$ |
| **MPLP($F = 1024$)** | $\mathbf{75.02_{\pm 2.68}}$ | $\mathbf{75.27_{\pm 2.95}}$ | $\mathbf{42.27_{\pm 3.96}}$ | $55.89_{\pm 3.03}$ |
| **MPLP($F = 2048$)** | $\mathbf{75.30_{\pm 2.14}}$ | $\mathbf{75.82_{\pm 2.15}}$ | $41.98_{\pm 3.21}$ | $\mathbf{57.11_{\pm 2.56}}$ |
| **MPLP($F = 4096$)** | $\mathbf{76.04_{\pm 1.57}}$ | $\mathbf{76.17_{\pm 2.04}}$ | $\mathbf{43.33_{\pm 2.93}}$ | $\mathbf{58.55_{\pm 2.47}}$ |

Table 13: Link prediction results on non-attributed benchmarks evaluated by Hits@20. The format is average score ± standard deviation. The top three models are colored by **First**, **Second**, **Third**.

| | USAir | NS | PB | Yeast | C.ele | Power | Router | E.coli |
|---|---|---|---|---|---|---|---|---|
| **CN** | $65.55_{\pm 4.10}$ | $74.00_{\pm 1.98}$ | $23.66_{\pm 3.11}$ | $60.44_{\pm 3.32}$ | $28.23_{\pm 8.47}$ | $11.57_{\pm 0.55}$ | $9.38_{\pm 1.05}$ | $47.46_{\pm 1.60}$ |
| **AA** | $74.66_{\pm 4.64}$ | $74.00_{\pm 1.98}$ | $24.65_{\pm 3.20}$ | $66.70_{\pm 2.73}$ | $38.74_{\pm 5.00}$ | $11.57_{\pm 0.55}$ | $9.38_{\pm 1.05}$ | $58.03_{\pm 2.93}$ |
| **RA** | $\mathbf{77.81_{\pm 3.34}}$ | $74.00_{\pm 1.98}$ | $22.66_{\pm 4.69}$ | $67.97_{\pm 2.48}$ | $39.74_{\pm 3.79}$ | $11.57_{\pm 0.55}$ | $9.38_{\pm 1.05}$ | $67.60_{\pm 1.99}$ |
| **GCN** | $61.95_{\pm 4.94}$ | $75.57_{\pm 4.25}$ | $23.60_{\pm 5.63}$ | $66.09_{\pm 2.96}$ | $23.89_{\pm 3.75}$ | $12.37_{\pm 3.20}$ | $19.35_{\pm 3.40}$ | $55.39_{\pm 9.43}$ |
| **SAGE** | $73.04_{\pm 4.03}$ | $48.92_{\pm 10.31}$ | $\mathbf{31.14_{\pm 2.27}}$ | $62.42_{\pm 6.51}$ | $36.25_{\pm 3.43}$ | $3.36_{\pm 0.80}$ | $25.01_{\pm 6.42}$ | $66.81_{\pm 4.82}$ |
| **SEAL** | $\mathbf{76.68_{\pm 6.32}}$ | $\mathbf{81.33_{\pm 3.69}}$ | $27.97_{\pm 1.63}$ | $\mathbf{76.50_{\pm 2.30}}$ | $\mathbf{41.33_{\pm 5.26}}$ | $\mathbf{26.56_{\pm 1.62}}$ | $\mathbf{46.80_{\pm 12.24}}$ | $\mathbf{74.21_{\pm 3.24}}$ |
| **Neo-GNN** | $74.14_{\pm 4.49}$ | $80.97_{\pm 2.73}$ | $\mathbf{28.96_{\pm 2.51}}$ | $\mathbf{75.35_{\pm 3.38}}$ | $38.18_{\pm 3.88}$ | $19.53_{\pm 5.53}$ | $31.94_{\pm 4.39}$ | $65.42_{\pm 4.07}$ |
| **ELPH** | $74.21_{\pm 4.94}$ | $\mathbf{85.42_{\pm 2.19}}$ | $26.88_{\pm 4.18}$ | $68.13_{\pm 3.76}$ | $\mathbf{41.61_{\pm 4.68}}$ | $\mathbf{22.00_{\pm 1.78}}$ | $\mathbf{48.97_{\pm 3.60}}$ | $66.48_{\pm 2.30}$ |
| **NCNC** | $74.40_{\pm 3.70}$ | $80.95_{\pm 2.46}$ | $28.47_{\pm 3.76}$ | $72.48_{\pm 2.47}$ | $36.15_{\pm 5.22}$ | $19.86_{\pm 1.05}$ | $37.23_{\pm 10.52}$ | $\mathbf{74.85_{\pm 4.58}}$ |
| **MPLP** | $\mathbf{83.67_{\pm 2.75}}$ | $\mathbf{83.67_{\pm 3.66}}$ | $\mathbf{33.69_{\pm 2.64}}$ | $\mathbf{81.95_{\pm 1.20}}$ | $\mathbf{50.07_{\pm 3.27}}$ | $\mathbf{28.70_{\pm 1.32}}$ | $44.04_{\pm 5.08}$ | $\mathbf{80.37_{\pm 1.89}}$ |

Table 14: Link prediction results on attributed benchmarks evaluated by Hits@20. The format is average score ± standard deviation. The top three models are colored by **First**, **Second**, **Third**.

| | CS | Physics | Computers | Photo | Collab |
|---|---|---|---|---|---|
| **CN** | $38.86_{\pm 0.28}$ | $44.17_{\pm 0.13}$ | $12.92_{\pm 1.96}$ | $18.97_{\pm 3.02}$ | $49.98_{\pm 0.00}$ |
| **AA** | $57.94_{\pm 3.66}$ | $58.27_{\pm 3.35}$ | $14.13_{\pm 2.45}$ | $23.18_{\pm 3.70}$ | $55.79_{\pm 0.00}$ |
| **RA** | $57.97_{\pm 2.72}$ | $56.12_{\pm 3.65}$ | $14.02_{\pm 1.57}$ | $24.21_{\pm 5.37}$ | $55.01_{\pm 0.00}$ |
| **GCN** | $50.13_{\pm 5.89}$ | $56.58_{\pm 3.48}$ | $14.63_{\pm 7.09}$ | $16.69_{\pm 4.88}$ | $24.39_{\pm 1.37}$ |
| **SAGE** | $44.50_{\pm 19.37}$ | $\mathbf{61.32_{\pm 3.95}}$ | $\mathbf{21.92_{\pm 2.88}}$ | $\mathbf{31.71_{\pm 3.62}}$ | $18.74_{\pm 4.64}$ |
| **SEAL** | $54.46_{\pm 2.04}$ | $57.66_{\pm 3.13}$ | $16.81_{\pm 0.90}$ | $27.13_{\pm 2.23}$ | $54.66_{\pm 0.91}$ |
| **Neo-GNN** | $\mathbf{58.17_{\pm 4.06}}$ | $58.64_{\pm 3.30}$ | $14.84_{\pm 1.19}$ | $28.00_{\pm 2.86}$ | $49.73_{\pm 0.82}$ |
| **ELPH** | $55.67_{\pm 2.87}$ | $48.05_{\pm 3.17}$ | $17.33_{\pm 2.73}$ | $27.18_{\pm 2.54}$ | $\mathbf{59.92_{\pm 0.19}}$ |
| **NCNC** | $\mathbf{59.99_{\pm 3.61}}$ | $\mathbf{60.80_{\pm 5.71}}$ | $\mathbf{22.40_{\pm 3.82}}$ | $\mathbf{29.08_{\pm 6.52}}$ | $\mathbf{56.89_{\pm 4.40}}$ |
| **MPLP** | $\mathbf{62.41_{\pm 3.07}}$ | $\mathbf{60.43_{\pm 3.92}}$ | $\mathbf{25.58_{\pm 3.76}}$ | $\mathbf{37.53_{\pm 3.18}}$ | $\mathbf{56.69_{\pm 1.15}}$ |

Table 15: Link prediction results on non-attributed benchmarks evaluated by Hits@100. The format is average score ± standard deviation. The top three models are colored by **First**, **Second**, **Third**.

| | USAir | NS | PB | Yeast | C.ele | Power | Router | E.coli |
|---|---|---|---|---|---|---|---|---|
| **CN** | $84.31_{\pm 4.21}$ | $74.00_{\pm 1.98}$ | $49.15_{\pm 3.87}$ | $73.76_{\pm 0.86}$ | $56.69_{\pm 1.55}$ | $11.57_{\pm 0.55}$ | $9.38_{\pm 1.05}$ | $58.00_{\pm 1.48}$ |
| **AA** | $90.80_{\pm 1.67}$ | $74.00_{\pm 1.98}$ | $53.07_{\pm 3.30}$ | $73.76_{\pm 0.86}$ | $75.66_{\pm 2.24}$ | $11.57_{\pm 0.55}$ | $9.38_{\pm 1.05}$ | $74.83_{\pm 1.48}$ |
| **RA** | $90.80_{\pm 1.67}$ | $74.00_{\pm 1.98}$ | $53.91_{\pm 3.67}$ | $73.76_{\pm 0.86}$ | $75.76_{\pm 2.12}$ | $11.57_{\pm 0.55}$ | $9.38_{\pm 1.05}$ | $78.70_{\pm 0.65}$ |
| **GCN** | $80.75_{\pm 3.86}$ | $80.29_{\pm 2.64}$ | $49.19_{\pm 4.35}$ | $77.13_{\pm 1.89}$ | $59.04_{\pm 4.34}$ | $20.52_{\pm 3.02}$ | $28.55_{\pm 5.88}$ | $64.78_{\pm 12.96}$ |
| **SAGE** | $90.92_{\pm 1.69}$ | $65.15_{\pm 7.99}$ | $\mathbf{61.00_{\pm 2.43}}$ | $77.34_{\pm 2.79}$ | $78.04_{\pm 4.32}$ | $11.51_{\pm 1.30}$ | $55.96_{\pm 4.52}$ | $80.45_{\pm 1.65}$ |
| **SEAL** | $\mathbf{95.74_{\pm 1.18}}$ | $\mathbf{91.02_{\pm 3.01}}$ | $59.87_{\pm 1.93}$ | $\mathbf{87.82_{\pm 0.66}}$ | $\mathbf{82.35_{\pm 3.11}}$ | $\mathbf{38.85_{\pm 2.65}}$ | $\mathbf{71.32_{\pm 4.97}}$ | $\mathbf{86.95_{\pm 0.67}}$ |
| **Neo-GNN** | $91.53_{\pm 1.63}$ | $85.29_{\pm 3.57}$ | $58.38_{\pm 2.67}$ | $84.98_{\pm 0.80}$ | $77.76_{\pm 3.05}$ | $26.46_{\pm 3.94}$ | $49.95_{\pm 4.54}$ | $78.75_{\pm 1.70}$ |
| **ELPH** | $\mathbf{94.52_{\pm 0.94}}$ | $\mathbf{92.01_{\pm 1.28}}$ | $\mathbf{61.11_{\pm 2.81}}$ | $\mathbf{85.92_{\pm 0.57}}$ | $\mathbf{80.70_{\pm 2.48}}$ | $\mathbf{33.49_{\pm 1.42}}$ | $\mathbf{69.26_{\pm 1.88}}$ | $80.04_{\pm 1.41}$ |
| **NCNC** | $91.13_{\pm 1.80}$ | $84.87_{\pm 3.76}$ | $59.34_{\pm 2.87}$ | $85.92_{\pm 0.78}$ | $76.64_{\pm 3.02}$ | $27.31_{\pm 2.39}$ | $63.93_{\pm 6.35}$ | $\mathbf{87.82_{\pm 0.65}}$ |
| **MPLP** | $\mathbf{95.41_{\pm 1.36}}$ | $\mathbf{92.01_{\pm 1.56}}$ | $\mathbf{65.98_{\pm 2.40}}$ | $86.78_{\pm 0.69}$ | $\mathbf{88.30_{\pm 2.09}}$ | $37.03_{\pm 1.12}$ | $69.79_{\pm 1.22}$ | $\mathbf{89.66_{\pm 0.68}}$ |

Table 16: Link prediction results on attributed benchmarks evaluated by Hits@100. The format is average score ± standard deviation. The top three models are colored by **First**, **Second**, **Third**.

|  | CS | Physics | Computers | Photo | Collab |
|---|---|---|---|---|---|
| **CN** | $69.05_{\pm 0.31}$ | $63.39_{\pm 0.14}$ | $29.58_{\pm 2.27}$ | $41.74_{\pm 1.87}$ | $65.60_{\pm 0.00}$ |
| **AA** | $69.05_{\pm 0.31}$ | $80.85_{\pm 0.80}$ | $38.02_{\pm 1.16}$ | $50.46_{\pm 2.33}$ | $65.60_{\pm 0.00}$ |
| **RA** | $69.05_{\pm 0.31}$ | $80.87_{\pm 0.91}$ | $41.74_{\pm 1.52}$ | $55.09_{\pm 2.98}$ | $65.60_{\pm 0.00}$ |
| **GCN** | $73.94_{\pm 1.69}$ | $\textbf{83.01}_{\pm 0.60}$ | $31.36_{\pm 13.66}$ | $38.15_{\pm 10.88}$ | $47.40_{\pm 2.08}$ |
| **SAGE** | $68.71_{\pm 14.24}$ | $82.59_{\pm 1.04}$ | $\textbf{45.79}_{\pm 4.06}$ | $59.44_{\pm 2.11}$ | $48.86_{\pm 5.19}$ |
| **SEAL** | $76.81_{\pm 0.88}$ | $81.77_{\pm 1.89}$ | $44.92_{\pm 1.08}$ | $\textbf{62.93}_{\pm 3.32}$ | $70.24_{\pm 0.25}$ |
| **Neo-GNN** | $76.43_{\pm 1.32}$ | $81.02_{\pm 0.80}$ | $34.38_{\pm 1.16}$ | $58.13_{\pm 2.92}$ | $62.34_{\pm 0.20}$ |
| **ELPH** | $79.03_{\pm 1.67}$ | $75.90_{\pm 1.97}$ | $41.40_{\pm 2.65}$ | $57.80_{\pm 2.52}$ | $70.43_{\pm 1.28}$ |
| **NCNC** | $81.32_{\pm 0.70}$ | $84.41_{\pm 0.90}$ | $49.38_{\pm 3.53}$ | $61.52_{\pm 2.48}$ | $71.87_{\pm 0.18}$ |
| **MPLP** | $82.25_{\pm 1.01}$ | $83.02_{\pm 1.02}$ | $53.92_{\pm 1.43}$ | $70.20_{\pm 2.56}$ | $71.55_{\pm 0.40}$ |

## E  THEORETICAL ANALYSIS

### E.1  PROOF FOR THEOREM 1

We begin by restating Theorem 1 and then proceed with its proof:

Let $G = (V, E)$ be a non-attributed graph and consider a 1-layer GCN/SAGE. Define the input vectors $\boldsymbol{X} \in \mathbb{R}^{N \times F}$ initialized randomly from a zero-mean distribution with standard deviation $\sigma_{node}$. Additionally, let the weight matrix $\boldsymbol{W} \in \mathbb{R}^{F' \times F}$ be initialized from a zero-mean distribution with standard deviation $\sigma_{weight}$. After performing message passing, for any pair of nodes $\{(u, v)|(u, v) \in V \times V \setminus E\}$, the expected value of their inner product is given by:

For GCN:

$$\mathbb{E}(\boldsymbol{h}_u \cdot \boldsymbol{h}_v) = \frac{C}{\sqrt{\hat{d}_u \hat{d}_v}} \sum_{k \in \mathcal{N}_u \bigcap \mathcal{N}_v} \frac{1}{\hat{d}_k},$$

For SAGE:

$$\mathbb{E}(\boldsymbol{h}_u \cdot \boldsymbol{h}_v) = \frac{C}{\sqrt{d_u d_v}} \sum_{k \in \mathcal{N}_u \bigcap \mathcal{N}_v} 1,$$

where $\hat{d}_v = d_v + 1$ and the constant $C$ is defined as $C = \sigma_{node}^2 \sigma_{weight}^2 F F'$.

*Proof.* Define $\boldsymbol{X}$ as $\left(\boldsymbol{X}_1^\top, \ldots, \boldsymbol{X}_N^\top\right)^\top$ and $\boldsymbol{W}$ as $(\boldsymbol{W}_1, \boldsymbol{W}_2, \ldots, \boldsymbol{W}_F)$.

Using GCN as the MPNN, the node representation is updated by:

$$\boldsymbol{h}_u = \boldsymbol{W} \sum_{k \in \mathcal{N}(u) \cup \{u\}} \frac{1}{\sqrt{\hat{d}_k \hat{d}_u}} \boldsymbol{X}_k,$$

where $\hat{d}_v = d_v + 1$.

For any two nodes $(u, v)$ from $\{(u, v) | (u, v) \in V \times V \setminus E\}$, we compute:

$$\boldsymbol{h}_u \cdot \boldsymbol{h}_v = \boldsymbol{h}_u^\top \boldsymbol{h}_v$$

$$= \left( \boldsymbol{W} \sum_{a \in \mathcal{N}(u) \cup \{u\}} \frac{1}{\sqrt{\hat{d}_a \hat{d}_u}} \boldsymbol{X}_a \right)^\top \left( \boldsymbol{W} \sum_{b \in \mathcal{N}(v) \cup \{v\}} \frac{1}{\sqrt{\hat{d}_b \hat{d}_v}} \boldsymbol{X}_b \right)$$

$$= \sum_{a \in \mathcal{N}(u) \cup \{u\}} \frac{1}{\sqrt{\hat{d}_a \hat{d}_u}} \boldsymbol{X}_a^\top \boldsymbol{W}^\top \boldsymbol{W} \sum_{b \in \mathcal{N}(v) \cup \{v\}} \frac{1}{\sqrt{\hat{d}_b \hat{d}_v}} \boldsymbol{X}_b$$

$$= \sum_{a \in \mathcal{N}(u) \cup \{u\}} \frac{1}{\sqrt{\hat{d}_a \hat{d}_u}} \boldsymbol{X}_a^\top \begin{pmatrix} \boldsymbol{W}_1^\top \boldsymbol{W}_1 & \cdots & \boldsymbol{W}_1^\top \boldsymbol{W}_F \\ \vdots & \vdots & \vdots \\ \boldsymbol{W}_F^\top \boldsymbol{W}_1 & \cdots & \boldsymbol{W}_F^\top \boldsymbol{W}_F \end{pmatrix} \sum_{b \in \mathcal{N}(v) \cup \{v\}} \frac{1}{\sqrt{\hat{d}_b \hat{d}_v}} \boldsymbol{X}_b.$$

Given that

1. $\mathbb{E}\big(\boldsymbol{W}_i^\top \boldsymbol{W}_j\big) = \sigma_{weight}^2 F'$ when $i = j$,

2. $\mathbb{E}\big(\boldsymbol{W}_i^\top \boldsymbol{W}_j\big) = 0$ when $i \neq j$,

we obtain:

$$\mathbb{E}(\boldsymbol{h}_u \cdot \boldsymbol{h}_v) = \sigma_{weight}^2 F' \sum_{a \in \mathcal{N}(u) \cup \{u\}} \frac{1}{\sqrt{\hat{d}_a \hat{d}_u}} \boldsymbol{X}_a^\top \sum_{b \in \mathcal{N}(v) \cup \{v\}} \frac{1}{\sqrt{\hat{d}_b \hat{d}_v}} \boldsymbol{X}_b.$$

Also the orthogonal of the random vectors guarantee that $\mathbb{E}\big(\boldsymbol{X}_a^\top \boldsymbol{X}_b\big) = 0$ when $a \neq b$. Then, we have:

$$\mathbb{E}(\boldsymbol{h}_u \cdot \boldsymbol{h}_v) = \frac{C}{\sqrt{\hat{d}_u \hat{d}_v}} \sum_{k \in \mathcal{N}_u \cap \mathcal{N}_v} \frac{1}{\hat{d}_k}$$

where $C = \sigma_{node}^2 \sigma_{weight}^2 F F'$.

This completes the proof for the GCN variant. A similar approach, utilizing the probabilistic orthogonality of the input vectors and weight matrix, can be employed to derive the expected value for SAGE as the MPNN. $\qquad \square$

### E.2 Proof for Theorem 2

We begin by restating Theorem 2 and then proceed with its proof:

Let $G = (V, E)$ be a graph, and let the vector dimension be given by $F \in \mathbb{N}_+$. Define the input vectors $\boldsymbol{X} = (X_{i,j})$, which are initialized from a random variable x having a mean of 0 and a standard deviation of $\frac{1}{\sqrt{F}}$. Using the message-passing as described by Equation 3, for any pair of nodes $\{(u, v) | (u, v) \in V \times V\}$, the expected value and variance of their inner product are:

$$\mathbb{E}(\boldsymbol{h}_u \cdot \boldsymbol{h}_v) = \mathrm{CN}(u, v),$$

$$\mathrm{Var}(\boldsymbol{h}_u \cdot \boldsymbol{h}_v) = \frac{1}{F} \big( d_u d_v + \mathrm{CN}(u, v)^2 - 2\mathrm{CN}(u, v) \big) + F \mathrm{Var}\big(\mathrm{x}^2\big) \mathrm{CN}(u, v).$$

*Proof.* We follow the proof of the theorem in Nunes et al. (2023). Based on the message-passing defined in Equation 3:

$$\mathbb{E}(\boldsymbol{h}_u \cdot \boldsymbol{h}_v) = \mathbb{E}\left( \left( \sum_{k_u \in \mathcal{N}_u} \boldsymbol{X}_{k_u,:} \right) \cdot \left( \sum_{k_v \in \mathcal{N}_v} \boldsymbol{X}_{k_v,:} \right) \right)$$

$$= \mathbb{E}\left( \sum_{k_u \in \mathcal{N}_u} \sum_{k_v \in \mathcal{N}_v} \boldsymbol{X}_{k_u,:} \boldsymbol{X}_{k_v,:} \right)$$

$$= \sum_{k_u \in \mathcal{N}_u} \sum_{k_v \in \mathcal{N}_v} \mathbb{E}(\boldsymbol{X}_{k_u,:} \boldsymbol{X}_{k_v,:}).$$

Since the sampling of each dimension is independent of each other, we get:

$$\mathbb{E}(\boldsymbol{h}_u \cdot \boldsymbol{h}_v) = \sum_{k_u \in \mathcal{N}_u} \sum_{k_v \in \mathcal{N}_v} \sum_{i=1}^{F} \mathbb{E}(X_{k_u,i} X_{k_v,i}).$$

When $k_u = k_v$,

$$\mathbb{E}(X_{k_u,i} X_{k_v,i}) = \mathbb{E}(\mathrm{x}^2) = \frac{1}{F}.$$

When $k_u \neq k_v$,

$$\mathbb{E}(X_{k_u,i} X_{k_v,i}) = \mathbb{E}(X_{k_u,i})\mathbb{E}(X_{k_v,i}) = 0.$$

Thus:

$$\mathbb{E}(\boldsymbol{h}_u \cdot \boldsymbol{h}_v) = \sum_{k_u \in \mathcal{N}_u} \sum_{k_v \in \mathcal{N}_v} \sum_{i=1}^{F} \mathbf{1}(k_u = k_v) \frac{1}{F}$$

$$= \sum_{k \in \mathcal{N}_u \cap \mathcal{N}_v} 1 = \mathrm{CN}(u,v).$$

For the variance, we separate the equal from the non-equal pairs of $k_u$ and $k_v$. Note that there is no covariance between the equal pairs and the non-equal pairs due to the independence:

$$\mathrm{Var}(\boldsymbol{h}_u \cdot \boldsymbol{h}_v) = \mathrm{Var}\left( \sum_{k_u \in \mathcal{N}_u} \sum_{k_v \in \mathcal{N}_v} \sum_{i=1}^{F} X_{k_u,i} X_{k_v,i} \right)$$

$$= \sum_{i=1}^{F} \mathrm{Var}\left( \sum_{k_u \in \mathcal{N}_u} \sum_{k_v \in \mathcal{N}_v} X_{k_u,i} X_{k_v,i} \right)$$

$$= \sum_{i=1}^{F} \left( \mathrm{Var}\left( \sum_{k \in \mathcal{N}_u \cap \mathcal{N}_v} \mathrm{x}^2 \right) + \mathrm{Var}\left( \sum_{k_u \in \mathcal{N}_u} \sum_{k_v \in \mathcal{N}_v \setminus \{k_u\}} X_{k_u,i} X_{k_v,i} \right) \right).$$

For the first term, we can obtain:

$$\mathrm{Var}\left( \sum_{k \in \mathcal{N}_u \cap \mathcal{N}_v} \mathrm{x}^2 \right) = \mathrm{Var}(\mathrm{x}^2)\mathrm{CN}(u,v).$$

For the second term, we further split the variance of linear combinations to the linear combinations of variances and covariances:

$$\mathrm{Var}\left( \sum_{k_u \in \mathcal{N}_u} \sum_{k_v \in \mathcal{N}_v \setminus \{k_u\}} X_{k_u,i} X_{k_v,i} \right) = \sum_{k_u \in \mathcal{N}_u} \sum_{k_v \in \mathcal{N}_v \setminus \{k_u\}} \mathrm{Var}(X_{k_u,i} X_{k_v,i}) +$$

$$\sum_{a \in \mathcal{N}_u \setminus \{k_u\}} \sum_{b \in \mathcal{N}_v \setminus \{k_v, a\}} \mathrm{Cov}(X_{k_u,i} X_{k_v,i}, X_{a,i} X_{b,i}).$$

Note that the $\mathrm{Cov}(X_{k_u,i} X_{k_v,i}, X_{a,i} X_{b,i})$ is $\mathrm{Var}(X_{k_u,i} X_{k_v,i}) = \frac{1}{F^2}$ when $(k_u, k_v) = (b, a)$, and otherwise 0.

Thus, we have:

$$\mathrm{Var}\left( \sum_{k_u \in \mathcal{N}_u} \sum_{k_v \in \mathcal{N}_v \setminus \{k_u\}} X_{k_u,i} X_{k_v,i} \right) = \frac{1}{F^2}\left( d_u d_v + \mathrm{CN}(u,v)^2 - 2\mathrm{CN}(u,v) \right),$$

and the variance is:

$$\mathrm{Var}(\boldsymbol{h}_u \cdot \boldsymbol{h}_v) = \frac{1}{F}\left( d_u d_v + \mathrm{CN}(u,v)^2 - 2\mathrm{CN}(u,v) \right) + F\mathrm{Var}(\mathrm{x}^2)\mathrm{CN}(u,v).$$

$\square$

E.3 PROOF FOR THEOREM 3

We begin by restating Theorem 3 and then proceed with its proof:

Under the conditions defined in Theorem 2, let $\boldsymbol{h}_u^{(l)}$ denote the vector for node $u$ after the $l$-th message-passing iteration. We have:

$$\mathbb{E}\left(\boldsymbol{h}_u^{(p)} \cdot \boldsymbol{h}_v^{(q)}\right) = \sum_{k \in V} |\text{walks}^{(p)}(k, u)||\text{walks}^{(q)}(k, v)|,$$

where $|\text{walks}^{(l)}(u, v)|$ counts the number of length-$l$ walks between nodes $u$ and $v$.

*Proof.* Reinterpreting the message-passing described in Equation 3, we can equivalently express it as:

$$\text{ms}_v^{(l+1)} = \bigcup_{u \in \mathcal{N}_v} \text{ms}_u^{(l)}, \boldsymbol{h}_v^{(l+1)} = \sum_{u \in \text{ms}_v^{(l+1)}} \boldsymbol{h}_u^{(0)}, \tag{9}$$

where $\text{ms}_v^{(l)}$ refers to a multiset, a union of multisets from its neighbors. Initially, $\text{ms}_v^{(0)} = \{\{v\}\}$. The node vector $\boldsymbol{h}_v^{(l)}$ is derived by summing the initial QO vectors of the multiset's elements.

We proceed by induction: Base Case ($l = 1$):

$$\text{ms}_v^{(1)} = \bigcup_{u \in \mathcal{N}_v} \text{ms}_u^{(0)} = \bigcup_{u \in \mathcal{N}_v} \{\{u\}\} = \{\{k | \omega \in \text{walks}^{(1)}(k, v)\}\}$$

Inductive Step ($l \geq 1$): Let's assume that $\text{ms}_v^{(l)} = \{\{k | \omega \in \text{walks}^{(l)}(k, v)\}\}$ holds true for an arbitrary $l$. Utilizing Equation 9 and the inductive hypothesis, we deduce:

$$\text{ms}_v^{(l+1)} = \bigcup_{u \in \mathcal{N}_v} \{\{k | \omega \in \text{walks}^{(l)}(k, u)\}\}.$$

If $k$ initiates the $l$-length walks terminating at $v$ and if $v$ is adjacent to $u$, then $k$ must similarly initiate the $l$-length walks terminating at $u$. This consolidates our inductive premise.

With the induction established:

$$\mathbb{E}\left(\boldsymbol{h}_u^{(p)} \cdot \boldsymbol{h}_v^{(q)}\right) = \mathbb{E}\left(\sum_{k_u \in \text{ms}_u^{(p)}} \boldsymbol{h}_{k_u}^{(0)} \cdot \sum_{k_v \in \text{ms}_v^{(q)}} \boldsymbol{h}_{k_v}^{(0)}\right)$$

The inherent independence among node vectors concludes the proof. $\square$

## F LIMITATIONS

Despite the promising capabilities of MPLP, there are distinct limitations that warrant attention:

1. Training cost vs. inference cost: The computational cost during training significantly outweighs that of inference. This arises from the necessity to remove shortcut edges for positive links in the training phase, causing the graph structure to change across different batches. This, in turn, mandates a repeated computation of the shortest-path neighborhood. A potential remedy is to consider only a subset of links in the graph as positive instances and mask them, enabling a single round of preprocessing. Exploring this approach will be the focus of future work.

2. Estimation variance influenced by graph structure: The structure of the graph itself can magnify the variance of our estimations. Specifically, in dense graphs or those with a high concentration of hubs, the variance can become substantial, thereby compromising the accuracy of structural feature estimation.

3. Optimality of estimating structural features: Our research demonstrates the feasibility of using message-passing to derive structural features. However, its optimality remains undetermined. Message-passing, by nature, involves matrix multiplication operations, which can pose challenges in terms of computational time and space, particularly for exceedingly large graphs.

