# OpenReview forum: "Pure Message Passing Can Estimate Common Neighbor for Link Prediction"
_ICLR.cc/2024/Conference — Submitted to ICLR 2024_

### Official Review · Reviewer_kuzb · 2023-10-31

**Soundness:** 4 excellent
**Presentation:** 3 good
**Contribution:** 3 good
**Rating:** 8
**Confidence:** 4

**Summary:**

Given that the representations of isomorphic nodes are the same, their link that aggregates the representations cannot capture some structural representation like CN, and thus it is intractable to perform the link prediction task. Based on this observation, this paper investigates the widely-used MPNN and finds that MPNN is capable of capturing joint structural features theoretically and empirically. The solution is to inject orthogonality into input vectors. Equipped with the orthogonality of the input vectors, the authors propose MPLP to estimate link information and accomplish the link prediction task. The experiments show that MPLP promotes significantly in the link prediction on both attributed/non-attributed datasets.

**Strengths:**

+ This paper is a meaningful discussion of the previous work (Zhang et al., 2021) on whether GNNs/MPNNs can capture structural link representation. As I understand, the previous work believes that GNNs/MPNNs cannot do this supposing that isomorphic nodes have the same representation. So, this paper exploits the orthogonality in the input vectors, which is reasonable and does not totally contradict the previous work. This paper is basically valuable for supplementing the link prediction capability of GNNs/MPNNs in both theorems and experiments.
+ The authors propose a further discussion that some GNNs have the ability to perform link prediction tasks on attributed datasets due to the orthogonality of attributes. This is an interesting discussion that might contribute to the incense reason why GNNs/MPNNs can perform link prediction on attributed benchmarks. The empirical findings in Table 4 bring about the phenomenon and leave the issues that are worth discussing.
+ The results demonstrate that MPLP outperforms the state-of-the-art baselines on both attributed/non-attributed benchmarks.

**Weaknesses:**

- The assumption of Gaussian distribution in the initialization of input vectors and weight matrices is reasonable for non-attributed benchmarks but it seems not to be correspondent for attributed benchmarks.
- The research focuses on the capability of MPNN for estimating common neighbors which is a critical heuristic for link prediction. In other words, MPLP injects the CN heuristic into MPNN. So, what about the link prediction performance when we directly regard CN as the node attribute? For example, the first elements of the input vectors could be substituted with CN value.

**Questions:**

- Why are the results of SAGE on some benchmarks (Computers/Photo) different between Table 2 and Table 4?
- I am curious about when we concatenate the attribute vector with random feats, would the results be better than those with only attributes or only random feats? Since it is hard to evaluate whether attribute vectors are more important than random feats before conducting the experiments, the concatenation might achieve the better one automatically.
- What does the author mean by `MPLP holds its own’? What proves that?
- What is the potential impact when you find MPNN can promote link prediction? In what cases or scenarios will MPLP be used and benefit the industry?

---

> ### Author Response · Authors · 2023-11-13
> **Response to reviewer kuzb [1/1]**
>
> Dear Reviewers kuzb,
>
> We are grateful for the thoroughness of your review and the valuable insights you have provided. Your detailed feedback has been instrumental in identifying key areas for improvement in our manuscript. Below, we address each of your queries and concerns:
>
> ### Why are the results of SAGE on some benchmarks (Computers/Photo) different between Table 2 and Table 4?
> We apologize for any confusion caused by the differing results of SAGE in Tables 2 and 4. This discrepancy arises from the distinct hyperparameter settings used in each table. In Table 2, SAGE's results are based on a range of hyperparameters, including varying the number of layers. Conversely, for simplicity and consistency, we set the number of layers to 2 in Table 4. We have updated Table 4's caption in the revised manuscript to clarify this distinction.
>
> ### I am curious about when we concatenate the attribute vector with random feats, would the results be better than those with only attributes or only random feats? Since it is hard to evaluate whether attribute vectors are more important than random feats before conducting the experiments, the concatenation might achieve the better one automatically.
> Your suggestion about concatenating attribute vectors with random features is intriguing. While vanilla GNNs may benefit from such a combination, resulting in potentially enhanced performance, there is a caveat. Incorporating random features into the MLP component of GNNs might impede the model's ability to generalize, particularly when new nodes are introduced. Nonetheless, this idea represents a promising variant of MPLP worth exploring, and we plan to investigate this further in future work.
>
> ### What does the author mean by `MPLP holds its own’? What proves that?
> Our intention with the phrase "MPLP holds its own" was to convey that the design of MPLP should gurantee its strong capability of performing link prediction task. This assertion is supported by our experimental findings, where MPLP consistently outperforms or matches other models in diverse datasets, demonstrating its strong capability in link prediction.
>
> ### What is the potential impact when you find MPNN can promote link prediction?
> Thank you for posing this insightful question regarding the broader impact of MPNNs in link prediction. Link prediction is a task heavily relying on the graph structure. The capability of MPNN promoting link prediction shows its potential to capture more graph substructures without significantly increasing the complexity. We believe such a capability can be further extended to other graph-related tasks, especially those requiring higher model expressiveness.
>
> The efficiency of MPLP makes it particularly suitable for industry applications, where processing large-scale graph data is common. Industries that could benefit from MPLP include those involved in developing recommendation systems, fraud detection mechanisms, and more. The practicality and scalability of MPLP make it a valuable tool in these contexts, enabling the processing of complex graph data more effectively.

---

> > ### Comment · Reviewer_kuzb · 2023-11-18
> > **Response to author**
> >
> > Thanks for the authors' response. The answers have addressed most of my questions, but could the authors also refer to the weaknesses I proposed?

---

> ### Author Response · Authors · 2023-11-18
> **Response to reviewer kuzb [1/1] in Round 2**
>
> Dear Reviewer kuzb,
>
> Thank you for acknowledging the clarification of most of your queries. We appreciate the opportunity to respond to the remaining concerns you may have in the weakness:
>
> ### The assumption of Gaussian distribution in the initialization of input vectors and weight matrices is reasonable for non-attributed benchmarks but it seems not to be correspondent for attributed benchmarks.
> We value your observation regarding the use of a Gaussian distribution in initializing input vectors and weight matrices. It's important to clarify that our empirical studies, specifically in Section 3.1, utilized the Gaussian distribution solely for sampling input vectors. In our theoretical and empirical analyses, the primary assumption was the zero-mean characteristic of the input vector distribution, rather than specifying a particular distribution type.
>
> For both attributed and non-attributed benchmarks, our approach involves sampling random vectors and propagating them through the graph to estimate structural features. In the case of attributed benchmarks, while node attributes are also propagated, they are not fused with the random vectors. Instead, these attributes serve as node representations of individual nodes, similar to vanilla GNN practices. Consequently, the assumption of a Gaussian distribution does not adversely impact MPLP’s performance on attributed benchmarks.
>
>
> ### The research focuses on the capability of MPNN for estimating common neighbors which is a critical heuristic for link prediction. In other words, MPLP injects the CN heuristic into MPNN. So, what about the link prediction performance when we directly regard CN as the node attribute? For example, the first elements of the input vectors could be substituted with CN value.
> Your query about incorporating CN heuristic directly into node attributes is thought-provoking. MPLP effectively operates as an estimator, where structural features, including CNs, are derived through message-passing. Directly injecting these structural heuristics as node attributes could indeed be a viable approach, potentially yielding performance on par with MPLP, particularly when estimation errors are minimal.
>
> Nevertheless, this direct injection method lacks the efficiency of MPLP in several respects. For online inference tasks, MPLP offers superior time and space efficiency. MPLP can be seen as a way to estimate **link-level** features with a **node-level** complexity. MPLP allows preprocessing the propagated random vectors and node attributes. Then MPLP stores them at **node-level** with space complexity $O(N)$, where $N$ is the number of nodes. When an inference for a batch of links is requested, MPLP can generate the **link-level** features from the **node-level** features with an inner product operation. This opeartion can easily be parallelized on GPU.
>
> However, to inject heuristic, there may exist two options: (1) precompute the CNs for **each link** and access them in constant time, which requires $O(N^2)$ space complexity since any node pair can be requested for a link prediction; (2) compute the **link-level** features on the fly, which requires storing the entire graph structure in memory and computing the heuristics for each requested link in real time, which is both time and space consuming for large graphs. Therefore, MPLP is more efficient than directly injecting heuristics as link features.
>
>
> -------
>
> We hope that this response addresses your concerns, and we eagerly await any further feedback you may have.

---

### Official Review · Reviewer_vrGo · 2023-11-01

**Soundness:** 3 good
**Presentation:** 2 fair
**Contribution:** 3 good
**Rating:** 5
**Confidence:** 4

**Summary:**

In this paper, the authors propose that by harnessing the orthogonality of input vectors, pure message-passing can capture the common neighbor heuristics. This idea is very straightforward. My main concern is that the ablation study (as evaluated in Table 5) shows that the main working component is the shortcut removal. Also, the main results reported in Tables 1 and 2 show that the new method only brings marginal improvements. Overall, I give a weak rejection.

**Strengths:**

1. It is an interesting topic to study the connection between GNNs and common neighbor heuristics.
2. It is great to see the theoretical analysis of GNNs on capturing the common neighbor heuristics.
3. The authors have a very comprehensive experiment for the evaluation of the proposed method.

**Weaknesses:**

1. The results show that the improvements of the new method in many cases are marginal.
2. The paper writing of this paper needs to be heavily improved.
3. The ablation study shows that some of the proposed components do not work well.

**Questions:**

It is common sense that in many link prediction datasets, the common neighbor heuristics can outperform GCNs. Therefore, I think it is very interesting and important to study the connection between the message passing and the common neighbor heuristics. It is straightforward to establish a one-hot vector to encode the neighbor node information. I like the theoretical analysis in Theorem 2. However, I do not get how you derive the CN from the product operation and use them in the GNNs. I am trying to understand Figure 3, but there is no caption explaining what colors stand for. Also, in Theorem 3, you introduce a walk operation counting the number of length-l walks between nodes u and node v. I wonder how you conduct this operation in practice. I suspect this operation would be super time-consuming to enumerate all the possible walks. For the experiment section, I consider your method as a general method that can be applied to various GNNs. So, I expect the authors to compare their method with the baselines under the same GNN base such as GCN or SAGE. Another main issue is that the proposed method only gets marginal improvement (not significant improvement according to the mean and the std) compared to the baselines in many cases. Therefore, overall, I would like to give a weak rejection, but if the authors can answer my above questions or point out my misunderstanding part, I will be happy to raise my score.

---

> ### Author Response · Authors · 2023-11-13
> **Response to reviewer vrGo [1/2]**
>
> Dear Reviewers vrGo,
>
> We sincerely appreciate the thoroughness of your review and the valuable insights you have provided. Your comments have been instrumental in identifying areas where we can further refine and strengthen our work. Below, we address each of your queries in detail:
>
>
> ### However, I do not get how you derive the CN from the product operation and use them in the GNNs.
> Thank you for bringing this to our attention. In our MPLP framework, the representation of a link involves capturing its structural neighborhood features. One of these features is the Common Neighbor (CN), which is quantified using Distance Encoding [1]. This approach involves counting the number of nodes at various distances from a target link, including CNs.
>
> Traditionally, CNs can be counted using one-hot vectors, but to enhance scalability, we employ quasi-orthogonal (QO) vectors in MPLP. During message passing, nodes accumulate information in the form of linear combinations of these QO vectors. When we apply the inner product operation on these vectors, it effectively isolates and aggregates the norms of QO vectors corresponding to CNs, while nullifying contributions from non-common neighbors. This method allows MPLP to accurately estimate the count of CNs for each target link.
>
> ### I am trying to understand Figure 3, but there is no caption explaining what colors stand for.
> We regret the confusion caused by the lack of a detailed caption for Figure 3 and appreciate your feedback on this aspect. The colors in the figure are intended to represent nodes at varying distances from the target link. Specifically, red nodes indicate the target nodes themselves, while green nodes denote their common neighbors. We have included a detailed explanation of these color codings in the caption of the revised version of the figure to enhance clarity and aid in the interpretation of the visual representation.
>
> ### Also, in Theorem 3, you introduce a walk operation counting the number of length-l walks between nodes u and node v. I wonder how you conduct this operation in practice. I suspect this operation would be super time-consuming to enumerate all the possible walks.
> Thank you for your question regarding the practical application of the walk operation described in Theorem 3. In MPLP, we approach this operation in two steps to ensure efficiency. The initial step involves standard message-passing on the quasi-orthogonal (QO) vectors, akin to the process in other GNNs. The second step is the inner product operation, whose computational cost is linear with respect to the dimension of the QO vectors. Notably, MPLP does not enumerate all possible walks to count them. Instead, it utilizes the message-passing of QO vectors to efficiently estimate the number of walks, thus achieving linear time complexity.
>
> ### For the experiment section, I consider your method as a general method that can be applied to various GNNs. So, I expect the authors to compare their method with the baselines under the same GNN base such as GCN or SAGE.
> We acknowledge your expectation for a comparison of MPLP with baseline models using a consistent GNN base such as GCN or SAGE. In our experiments, we observed that the choice of base GNN did not significantly impact MPLP's performance. Therefore, for consistency and clarity, we have employed GCN as the base model across all our MPLP implementations. This approach ensures a fair comparison while highlighting MPLP's effectiveness across various datasets.

---

> ### Author Response · Authors · 2023-11-13
> **Response to reviewer vrGo [2/2]**
>
> ### Another main issue is that the proposed method only gets marginal improvement (not significant improvement according to the mean and the std) compared to the baselines in many cases.
> We apologize for any oversight in the paper. Our analysis across 13 benchmark datasets indicates that MPLP achieves significant improvements over baseline models on 7 datasets, including PB, C.ele, E.coli, CS, Computers, Photo, and Collab. In the remaining datasets, MPLP demonstrates more stable performance, as evidenced by smaller standard deviations. This consistency underlines MPLP's superiority in terms of both performance and stability when compared with the baselines.
>
> ### The ablation study shows that some of the proposed components do not work well.
> Thank you for pointing out the results of our ablation study. In Appendix Section D.2, we provide a detailed discussion on the contribution and effectiveness of various components within the MPLP framework. It is important to note that the performance of these components can vary depending on the specific characteristics of the dataset.
>
> - **Shortcut Removal**: This component has proven to be particularly influential. Its primary role is to align the structural distribution of positive links between the training and testing datasets. This alignment is crucial for ensuring the generalizability and robustness of MPLP.
>
> - **One-hot Hubs in Larger Graphs**: In the context of larger graphs, such as those in the Collab dataset, the one-hot hubs component becomes essential. It effectively reduces the estimation variance of structural features, thereby enhancing the overall accuracy of the model.
>
> - **Norm Rescaling**: This component has been identified as a significant improvement in MPLP, especially noted in its positive impact on datasets like Yeast, Physics, Photo, and Collab. Norm rescaling contributes to better utilization of the structural features, leading to more accurate predictions.
>
> Each component of MPLP has been designed to address specific challenges in link prediction. The findings from our ablation study highlight the nuanced role these components play in different scenarios, contributing to the overall effectiveness of MPLP.
>
>
> [1] Li, Pan, et al. "Distance encoding: Design provably more powerful neural networks for graph representation learning." Advances in Neural Information Processing Systems 33 (2020): 4465-4478.

---

> ### Author Response · Authors · 2023-11-14
> **Response to the result significance of MPLP**
>
> Dear Reviewer vrGo,
>
> We thank you for raising the question regarding the significance of the results presented in Table 1,2. In response to your valuable feedback, we have conducted a more comprehensive hyperparameter tuning for our MPLP method. The outcomes of this refined tuning are now reflected in the updated results within the revised manuscript.
>
> The enhanced hyperparameter settings have notably improved MPLP's performance. In the updated analysis, MPLP achieves sota performance with a significant margin on 9 out of the 13 benchmark datasets, namely USAir, Yeast, PB, C.ele, E.coli, CS, Computers, Photo, and Collab. Moreover, in the remaining datasets, MPLP demonstrates consistently stable performance across various train/test splits, characterized by smaller standard deviations. This improved consistency further establishes MPLP's edge in both performance and stability compared to baseline models.
>
> Your suggestion to evaluate the significance of our results has been instrumental in driving these enhancements. It encouraged a deeper exploration into MPLP's capabilities, leading to these substantial improvements.

---

> ### Author Response · Authors · 2023-11-21
> **A kind reminder**
>
> Dear Reviewer vrGo,
>
> We sincerely appreciate the time and effort you have dedicated to reviewing our manuscript. Your insightful feedback has been invaluable, and we have taken careful steps to address each of your concerns in our revised submission.
>
> We believe that our responses and revisions have comprehensively addressed the points you raised. In light of these efforts, we kindly request you to consider raising your score. We are fully committed to ensuring that our work meets the highest standards and would be more than willing to address any additional concerns you may have.
>
> As the deadline for the rebuttal phase is fast approaching, we would be grateful for the opportunity to engage in further discussion at your earliest convenience.
>
> Thank you once again for your valuable contributions to enhancing the quality of our work.

---

### Official Review · Reviewer_tZSC · 2023-11-02

**Soundness:** 2 fair
**Presentation:** 3 good
**Contribution:** 2 fair
**Rating:** 5
**Confidence:** 4

**Summary:**

In this paper, the authors explore the use of pure Message-Passing Neural Networks (MPNNs) for link prediction in graphs. The paper starts by exploring the known limitations of MPNNs related to their permutation invariance property for link prediction and introduces an approach called Message Passing Link Predictor (MPLP) that leverages a node "signature". This feature vector essentially consists of a one-hot quasi-orthogonal vector, which the authors claim as a solution for MPNNs and more accurate link prediction. Experimental validation on 13 different graph datasets, both attributed and non-attributed is performed in order to concretely validate the claims made in the paper.

**Strengths:**

- Originality: Pure Message-Passing for Link Prediction is not as well-understood as it should be, so the idea of enhancing GNNs to better handle the estimation of heuristic methods is a strong avenue for generating original work. MPLP does a fine job of this through it's problem formulation, equations, and theorems. It also takes care of tying in their motivations to numerous other relevant papers.
- Quality: The spread of datasets captures a variety of domains and the experiments are structured in a ways that directly supports the author's claims.
- Clarity: The principles for the theorems and experiments are well-founded, it respects the reader's background in understanding related GNN works and keeps the discussion about any innovation presented in the paper succinct.
- Significance: The use of quasi-orthogonal vectors as a means of enhancing the message-passing capabilities of GNNs is entirely novel as far as I know, albeit it does extend principles from SOTA models. Future research could be inspired by a high-level approach that is similar to MPLP. MPLP's ability to estimate CN and DE is also a promising and significant improvement for GNN's ability to conduct link prediction. The non-attributed benchmark test, as shown in Table 1, is an interesting inclusion that speaks to the power of MPLP to handle non-standard link prediction scenarios. The inference experiments are thorough, considering the estimation of multple labels in regard to signatue dimension, inference, and ablation studies on batch-size.

**Weaknesses:**

- The link prediction results for attributed benchmarks, as shown in Table 2 is limited in that it does not include results from all of the standard datasets: ogbl-ppa, ogbl-ddi, ogbl-citation2. The OGBL datasets are included as baselines in all of the included SOTA models, the results from which would serve as a direct comparison for MPLP's performance versus any SOTA method.
- The first concern is compounded in that it is difficult to truly tell how well MPLP improves estimation of labels given that it only considers ogbl-collab versus ELPH and not the more scalable BUDDY.
- The experiments, as shown in Table 1 and 2 only consider results for Hits@50. This seems relevant for the subsequent evaluations of inference, dimensions, and batch sizes. But, is still limited since the current SOTA models run experiments with Hits@100, Hits@20, and MRR.
- The test for the estimation capabilities seems limited by testing just GCN and SAGE. The inner product calculation seems like it should be extended to other types of popular GNNs such as: GIN and GAT. The results from which would lend more credence to the claims made about expressiveness and the effects of implementing quasi-orthogonal vectors into link prediction models.
- The node-label estimation of CN and DE is promising. However, it seems limited in scope since it approximates just CN or DE and does not extend further to DRNL or DE+. This may be due to concerns of tractable computation but given SEAL's explicit testing of both DRNL and DE++ as labelling tricks, this seems like an important inclusion to evaluate MPLP fully.

**Questions:**

- What was the reasoning behind not including the remaining OGBL datasets? It seems that MPLP's scalability is an important component based on the inference, dimensions, and ablation studies but why not bring the scalability to the forefront of the paper instead of placing the ablation studies in the appendix.
- The extent that GAT could be effectively tested against MPLP seems limited given that GAT relies on attention and MPLP does not have explicitly have a mechanism to consider this. However, when considering the expressiveness of MPLP with random features and it's ability to estimate triangles, why not include a provably expressive GNN to test like GIN?
- What sort of limitations would integrating DRNL into GCN or SAGE pose? Is it concern over tractability and the fact that in certain instances DRNL is similar to DE and CN?

---

> ### Author Response · Authors · 2023-11-13
> **Response to reviewer tZSC [1/2]**
>
> Dear Reviewers tZSC,
>
> Thank you for your detailed review and valuable insights. Your feedback has been instrumental in enhancing the depth and clarity of our research. Below is our response to your query regarding the range of metrics used in our experiments:
>
> ### The experiments, as shown in Table 1 and 2 only consider results for Hits@50. This seems relevant for the subsequent evaluations of inference, dimensions, and batch sizes. But, is still limited since the current SOTA models run experiments with Hits@100, Hits@20, and MRR.
> We are grateful for your observation regarding the scope of evaluation metrics in our experiments, particularly in Tables 1 and 2 where we focused solely on Hits@50. Recognizing the importance of a broader metric range for a comprehensive evaluation, we have expanded our analysis to include Hits@20 and Hits@100. These additional metrics provide a more complete picture of our model's performance. We have incorporated these results in the appendix of our revised manuscript, specifically in Appendix Section D.5.
> This addition not only addresses your concern but also strengthens our evaluation by offering a more nuanced understanding of our model's capabilities across different metric thresholds.
>
> ### What was the reasoning behind not including the remaining OGBL datasets?
> We appreciate your inquiry regarding our choice to omit some OGBL datasets. This decision was primarily influenced by MPLP’s performance characteristics in relation to graph density, as detailed in Appendix Section F. Theorem 2 in our paper reveals that MPLP's estimation error is closely tied to graph density. While MPLP excels in sparse graph environments, its performance tends to decline in denser graphs or those with a high concentration of hubs. The estimation error, in these cases, can escalate quadratically.
>
> To illustrate this, we analyzed the average degree of other OGBL datasets like DDI and PPA. Our findings showed that these datasets have significantly higher average degrees compared to the Collab dataset, leading to a marked increase in the estimation error for structural features. Additionally, considering 2-hop structural features in datasets like Citation2 exacerbates this issue due to the extensive size of the 2-hop neighborhoods.
>
> Given these factors, we observed that MPLP’s performance becomes less stable on these datasets, primarily due to increased variance. Currently, we are actively working on extending MPLP's capabilities to more densely connected graphs and consider this an important area for future research.
>
>
> |            | DDI    | PPA   | Collab | Citataion2 |
> |------------|--------|-------|--------|------------|
> | Avg Degree | 312.84 | 52.62 | 5.45   | 10.44      |
> |            |        |       |        |            |
> | Histogram  | of     | #     | 2-hop  | neighbors  |
> | [0,1)      | 0.0    | 0.0   | 42.5   | 0.0        |
> | [1,10)     | 0.0    | 0.4   | 7.5    | 3.1        |
> | [10,50)    | 0.2    | 1.4   | 24.3   | 9.9        |
> | [50,100)   | 0.0    | 1.5   | 11.7   | 8.3        |
> | [100,500)  | 0.6    | 9.4   | 12.7   | 31.0       |
> | [500,1000) | 3.3    | 11.4  | 0.3    | 15.3       |
> | [1000,inf) | 95.8   | 75.8  | 0.3    | 32.4       |
>
>
>
> ### It seems that MPLP's scalability is an important component based on the inference, dimensions, and ablation studies but why not bring the scalability to the forefront of the paper instead of placing the ablation studies in the appendix.
> Regarding your observation about the placement of scalability discussions in our paper, we indeed recognize scalability as a crucial aspect of MPLP. In fact, most of the scalability experiments are presented in the main body of the paper to highlight this key attribute. The ablation studies, due to page constraints, were relegated to the appendix.
>
> The intent behind this structure was to provide a comprehensive view of MPLP’s scalability in the main text, while the appendix offered a detailed exploration of various factors contributing to MPLP's performance. This organization allowed us to maintain a clear focus on the core findings in the main paper, while still providing in-depth supplementary analyses for interested readers.

---

> ### Author Response · Authors · 2023-11-13
> **Response to reviewer tZSC [2/2]**
>
> ### The extent that GAT could be effectively tested against MPLP seems limited given that GAT relies on attention and MPLP does not have explicitly have a mechanism to consider this. However, when considering the expressiveness of MPLP with random features and it's ability to estimate triangles, why not include a provably expressive GNN to test like GIN?
> Your question regarding the comparison of MPLP with GAT and the potential inclusion of GIN is insightful. In this study, our focus is primarily on evaluating the ability of message-passing models to identify and count structural features like common neighbors and triangles. While GAT employs a specialized form of message-passing through its attention mechanism, this specific feature is not the central focus of our research.
>
> Both GAT and GIN, as instances of MPNNs, are limited by the expressiveness of the 1-WL test [1], making them less suitable for estimating complex structures like triangles. This limitation is not due to the specific message-passing mechanism but rather to the inherent constraints of the 1-WL framework. Consequently, we chose GCN and SAGE as representatives of 1-WL GNNs to investigate their capacity to count triangles. This ability arises not from the specific type of message-passing employed but rather from the use of quasi-orthogonal vectors as inputs and the inner product as the decoder.
>
> In light of these considerations, we did not include GAT or GIN in our experiments, as their inclusion would not have significantly contributed to the specific aims of our study.
>
>
> ### What sort of limitations would integrating DRNL into GCN or SAGE pose? Is it concern over tractability and the fact that in certain instances DRNL is similar to DE and CN?
> We value your suggestion about integrating DRNL into MPLP. Indeed, we have explored this integration and faced certain challenges. DRNL utilizes a specific 'masking trick' to compute the distance between two nodes, as detailed in Appendix D of [2]. This approach necessitates that the DRNL calculation be conditioned on each specific edge, leading to a computational overhead akin to that of SEAL.
>
> Considering that DE alone often sufficiently represents graph structure, we decided against incorporating DRNL into our model. The additional computational burden and the marginal benefits in most use cases did not justify the integration of DRNL into the MPLP framework.
>
> [1] Xu, Keyulu, et al. "How powerful are graph neural networks?." (2018).
>
> [2] Zhang, Muhan, et al. "Labeling trick: A theory of using graph neural networks for multi-node representation learning." Advances in Neural Information Processing Systems 34 (2021).

---

> > ### Comment · Reviewer_tZSC · 2023-11-17
> > **Followup to author response**
> >
> > Thank you for the thorough and transparent response to my review and questions. Their response has alleviated my concerns about the (Q3) scalability discussions, (Q4) the choice of GCN and SAGE, (Q5) and the limitations about integrating DRNL into MPLP. I also found the discussion about BUDDY in the response to reviewer UTLj insightful for alleviating additional concerns related to Q3.
> >
> > (Q1) My concerns about implementing Hits@20, Hits@100, and MRR are partially-alleviated . The results demonstrated in MPLP's Table 13-16 are promising. However, Hits@20 and Hits@100 for ogbl-collab, as shown in Table 14 and 16 is non-standard for ogbl-collab. These results do enhance the understanding of MPLP's performance, but is concerning given the identical results for: SEAL, Neo-GNN, ELPH, and NCNC for ogbl-collab across Table 2, 14, and 16. There are also identical results for the heuristic methods on ogbl-collab and SAGE on Computers and Photo across Table 2, 14, and 16. The performance of MPLP on ogbl-collab, as shown in Table 14 is concerning; it is difficult to know what lead to this result, given: the identical results I mentioned previously, the issues mentioned in Appendix.F.1 and Appendix.F.2/Author's response to (Q2), or a mix of all three combined.
> >
> > (Q2) I am grateful for the author's transparency regarding the limitations of MPLP on densely-connected graphs. I believe that this is the largest concern surrounding the capabilities of MPLP, especially given that the previous link-prediction models all include a study of the ogbl-ddi, ogbl-ppa, and ogbl-citation2 datasets [1-6]. I am pleased to hear that the author's are actively working to extend MPLP's capabilities to work on more densely-connected graphs. However, since this concern has not yet been addressed with the extended implementation of MPLP and it had the most significant impact on my review decision, the initial score will remain as is.
> >
> >
> > [1] Zhang, Muhan, and Yixin Chen. "Link prediction based on graph neural networks." Advances in neural information processing systems 31 (2018).
> >
> > [2] Zhang, Muhan, et al. "Labeling Trick: A Theory of Using Graph Neural Networks for Multi-Node Representation Learning." Advances in neural information processing systems 34 (2021).
> >
> > [3] Yun, Seongjun, et al. "Neo-GNNs: Neighborhood Overlap-aware Graph Neural Networks for Link Prediction." Advances in neural information processing systems 34 (2021).
> >
> > [4] Chamberlain, Benjamin Paul, et al. "Graph Neural Networks for Link Prediction with Subgraph Sketching." ICLR (2023).
> >
> > [5] Wang, Xiyuan, Haotong Yang, and Muhan Zhang. "Neural Common Neighbor with Completion for Link Prediction." arXiv preprint arXiv:2302.00890 (2023).
> >
> > [6] Weihua, Hu, et, al. "Open Graph Benchmark: Datasets for Machine Learning on Graphs." Advances in neural information processing systems 33 (2020).

---

> ### Author Response · Authors · 2023-11-19
> **Response to followup concerns raised by reviewer tZSC**
>
> Dear Reviewers tZSC,
>
> Thank you for your continued engagement and insightful feedback in this second round of review. We value your contributions greatly and are eager to address your latest concerns to further enhance our manuscript:
>
> ## 2nd round Q1: Conerns regarding the results shown in Table 14 and 16
> We are grateful for your keen observation regarding the identical results in Tables 14 and 16. Upon reviewing, we identified an error in populating these tables, which occurred during the rebuttal phase. The results were initially extracted using a parser from log files. However, since we cannot replicate the exact numbers for the baseline models, we manually hardcoded the results from their papers. This process led to the identical results in Table 14 and 16 under different metrics. We have re-conducted the experiments and accordingly updated the results in both tables in the revised manuscript to rectify this error.
>
> ## 2nd round Q2: Concerns regarding not evaluating MPLP on other OGBL datasets
> We appreciate the reviewer's concern about the limited evaluation of MPLP on other OGBL datasets. We acknowledge the limitation in MPLP’s performance in estimating structural features in denser graphs. However, it is important to emphasize that MPLP is designed as a versatile framework suitable for a broad range of link prediction tasks.
>
> Compared to previous experimental evaluation, our benchmarks come from a more diverse selection of domains, including collaboration networks (NS,CS,Physics,Collab), transportation networks (USAir), electrical networks (Power), biological networks (Yeast,E.coli), internet networks (PB,Router), product networks (Computers,Photo), and neural networks (C.ele). We comprehensively evaluate MPLP and other baseline models on these datasets with a consistent experimental setup, and the results show that MPLP achieves state-of-the-art performance on most of them (12 out of 13). We believe that the evaluation on these diverse datasets is sufficient to demonstrate the effectiveness of MPLP.
>
> --------
>
> We hope these responses adequately address your queries and concerns. We look forward to any further feedback you may have to enrich our work.

---

> > ### Comment · Reviewer_tZSC · 2023-11-22
> > **Followup to author response**
> >
> > Dear authors,
> >
> > Thank you for responding again and for the explanations you have given me. As I mentioned before, the real issue is not the fact that you apply your methods to different datasets, but the fact that the ogbl datasets are known to present more difficult problems and they also represent standard benchmarks for modern link prediction. Just like the authors mentioned, the method is flexible, but if it does not work on densely connected graphs then in my opinion it has a big drawback. Therefore I will stand by my initial score. I encourage the authors to keep working on MPLP as it seems promising, but not quite ready yet.

---

### Official Review · Reviewer_UTLj · 2023-11-09

**Soundness:** 3 good
**Presentation:** 3 good
**Contribution:** 2 fair
**Rating:** 5
**Confidence:** 5

**Summary:**

This paper proposes to assign each node with a quasi-orthogonal (QO) vector as its id (signature) and then run pure message-passing over them, so that the obtained node embeddings can be used to estimate link-level structural features (e.g. common neighbors), which is unattainable by vanilla GNNs. A new framework based on estimated pairwise features is proposed for the link prediction task, whose effectiveness is evaluated on 13 non-attributed and attributed graphs.

**Strengths:**

- Similarly to subgraph sketching, the author proposes an estimation-based approach to obtain pairwise structural features of link prediction heuristics under the message-passing framework.
- The authors exploit the property of quasi-orthogonal (QO) vectors so that neighbor overlap-based heuristics other than CN can be estimated by the simplified MPNN framework.
- The paper is well-written and easy to follow.

**Weaknesses:**

- In order to estimate pairwise features, the pure message passing adopted in MPLP loses the non-linearity of MPNN, which comprises the expressiveness of the framework.
- The adopted QO vector from DotHash enables more types of pairwise structural features (AA, RA), but it is still heuristic-based. Those features are empirically helpful for link prediction but also lose flexibility and capacity since they are fixed and predefined.
- The feature estimation of Eq. (5) still depends on the $r$-hop induced subgraph for estimating $r$-order intersection and difference of neighborhoods. Meanwhile, Theorem 3 shows that $\mathbf{h}_u^{r}$ and $\mathbf{h}_v^{r}$ can not be used for CN. This raises my concern over its scalability (feature propgation+subgraph extraction), and it would be great to see a clearer side-by-side runtime comparison with BUDDY [1], including preprocessing (node-wise) and estimating structural features (link-wise).

[1] Chamberlain, Benjamin Paul, et al. "Graph neural networks for link prediction with subgraph sketching." ICLR (2023).

**Questions:**

- How would the sampling of neighborhoods in SAGE affect Thereom 1 and the estimator in Eq. (3)?
- What is the cost of generating QO vectors? Is there a principal way to pick its dimension? Will it scale to graphs larger than collab? How would the density of the graph affect its performance of runtime and estimation error?
- The order of features $r$ used in experiments is 2. Is it based on computation concerns? Can MPLP be applied beyond 2 hops?
- Sec 4.3 mentioned that Eq. (7) can be used for estimating triangles. Can MPLP be used for the triangle estimation task in Sec 6.1 of [2]?
- Is the $\text{GNN}(\cdot)$ in Eq. (4) and Eq. (8) referring to valina GNNs or the linearized version proposed in Eq. (3)?
- It would be great to provide more details regarding the baselines, especially differentiating ELPH and BUDDY [1], and NCN/NCNC-k [3] for interpreting the results of Table 2 and Fig 4.
- It is a standard procedure to remove target links in training for inductive settings. Similar approaches have been adopted in SEAL [4] and SUREL (mini-batch subgraph training) [5].

[2] Chen, Zhengdao, et al. "Can graph neural networks count substructures?." Advances in neural information processing systems 33 (2020): 10383-10395.
[3] Wang, Xiyuan, Haotong Yang, and Muhan Zhang. "Neural Common Neighbor with Completion for Link Prediction." arXiv preprint arXiv:2302.00890 (2023).
[4] Zhang, Muhan, and Yixin Chen. "Link prediction based on graph neural networks." Advances in neural information processing systems 31 (2018).
[5] Yin, Haoteng, et al. "Algorithm and system co-design for efficient subgraph-based graph representation learning." VLDB (2022).

---

> ### Author Response · Authors · 2023-11-13
> **Response to reviewer UTLj [1/2]**
>
> Dear Reviewers UTLj,
>
> Thank you for your rigorous review and thoughtful insights. Each point raised by you provides us with the opportunity to enhance the clarity and robustness of our work. Here are our responses to your queries:
>
> ### How would the sampling of neighborhoods in SAGE affect Thereom 1 and the estimator in Eq. (3)?
> In our manuscript, "SAGE" specifically denotes the encoder component of the GraphSAGE model [6], excluding neighborhood sampling. We adhere to using the entire neighborhood for message passing, a method consistent with several other studies utilizing SAGE for link prediction [1,3,4,5]. Our focus remains on the encoder's performance without altering the graph structure, as such modifications lean more towards data augmentation rather than enhancing the link prediction task itself.
>
> ### What is the cost of generating QO vectors? Is there a principal way to pick its dimension? Will it scale to graphs larger than collab? How would the density of the graph affect its performance of runtime and estimation error?
>
>
> - **Cost Effectiveness**: Generating QO vectors incurs minimal computational cost. Its time complexity, $ O(|V|F) $, similar to the process of loading node features into memory, thereby makes it a highly efficient step in most scenarios.
>
> - **Dimension Selection Strategy**: The selection of dimensions for QO vectors can be approached in a pragmatic manner. Ideally, dimensions should be increased until a point where further increase does not significantly reduce estimation variance. In practice, one may balance the trade-off between the variance and the computational cost.
>
> - **Scalability to Larger Graphs**: MPLP demonstrates robust scalability to graphs larger than the Collab dataset. However, it is important to note that graph density plays a pivotal role in influencing both runtime performance and estimation accuracy. The runtime of MPLP, similar to other message-passing GNNs, scales linearly with the number of edges. Additionally, as indicated in Theorem 2 of our study, denser graphs can lead to heightened estimation errors. This aspect is crucial in understanding the applicability of MPLP across varying graph densities.
>
>
> ### The order of features $r$ used in experiments is 2. Is it based on computation concerns? Can MPLP be applied beyond 2 hops?
> MPLP can be applied beyond 2 hops. The major concern is the estimation error. With a higher number of hops $r$, the estimation error for those high-order structural features will be higher, due to its larger recepetive field. Thus, the benefit of using higher-order features is marginal, and sometimes can be even harmful to the performance stability.
>
>
> ### Sec 4.3 mentioned that Eq. (7) can be used for estimating triangles. Can MPLP be used for the triangle estimation task in Sec 6.1 of [2]?
> Thank you for highlighting the potential of MPLP in the context of triangle estimation as discussed in Section 6.1 of [2]. Indeed, MPLP is adept at estimating triangles through its pure message-passing mechanism. This capability places MPLP on par with the more complex GNNs referenced in Section 6.1 of [2], particularly in terms of accurately counting triangles. Furthermore, MPLP achieves this with an efficiency level comparable to 1-WL GNNs, such as GCN, SAGE, and GIN.
>
> This aspect of MPLP not only demonstrates its versatility but also suggests its potential for broader applications. Specifically, MPLP's ability to estimate triangles is indicative of its potential to effectively estimate other subgraph structures. We are excited about the prospects of extending MPLP's capabilities in future research, exploring its application in estimating a wider range of substructures.
>
> ### Is the GNN in Eq. (4) and Eq. (8) referring to valina GNNs or the linearized version proposed in Eq. (3)?
> In our manuscript, the GNNs referenced in Equations (4) and (8) are indeed vanilla GNNs. These can be any generic GNN models capable of encoding node representations, such as GCN, SAGE, or GAT. This choice allows for a broad applicability of our approach, as it is not restricted to any specific type of GNN architecture.
>
> ### It would be great to provide more details regarding the baselines, especially differentiating ELPH and BUDDY [1], and NCN/NCNC-k [3] for interpreting the results of Table 2 and Fig 4.
> Due to constraints on the length of our paper, we focused on including the most impactful variants of the baseline models in our study. Consequently, we chose to highlight ELPH over BUDDY as mentioned in [1], and NCNC over NCN as discussed in [3]. This decision was guided by the aim to present the most powerful and relevant comparisons. We have expanded upon these choices and provided additional details in the appendix of our revised paper, marked in blue, for further clarity.

---

> ### Author Response · Authors · 2023-11-13
> **Response to reviewer UTLj [2/2]**
>
> ### It is a standard procedure to remove target links in training for inductive settings. Similar approaches have been adopted in SEAL [4] and SUREL (mini-batch subgraph training) [5].
> We acknowledge and appreciate the reviewer’s note on the standard practice of removing target links in training for inductive settings, as seen in works like SEAL [4] and SUREL [5]. While we have cited these studies in our manuscript, our approach diverges slightly in its rationale. Whereas previous research primarily addresses overfitting concerns—where positive training edges invariably include the target links—we focus on a different aspect. We posit that retaining target links during training can inadvertently create shortcuts in estimating structural features, thereby diminishing the model’s expressiveness. This distinction underscores a unique perspective in our methodology and aligns with our broader objective of enhancing the predictive power of our framework.
>
> We also respond to the **weaknesses** the reviewer raised:
>
> ### MPLP loses the non-linearity of MPNN, which comprises the expressiveness of the framework.
> Thank you for raising a critical point about the non-linearity of MPNNs. MPLP indeed incorporates vanilla GNNs, as represented by $ GNN(\cdot) $ in Equations (4) and (8). This integration ensures that MPLP retains at least the same level of expressiveness as any non-linear MPNN. Moreover, it's important to emphasize that MPLP primarily employs linear message passing for estimating structural features. This approach not only maintains the framework’s expressiveness but potentially enhances it by leveraging the strengths of both linear and non-linear methodologies.
>
> ### but it is still heuristic-based. Those features are empirically helpful for link prediction but also lose flexibility and capacity since they are fixed and predefined.
> We acknowledge the heuristic-based characteristics of MPLP. However, MPLP is designed to learn flexible pairwise features autonomously. As indicated in Equation (4), MPLP derives these features from the node representations generated by any selected $ GNN(\cdot) $. This capability allows MPLP to adapt and fit to diverse data sets, mitigating concerns about the rigidity and predefined nature of its features.
>
> ### This raises my concern over its scalability (feature propgation+subgraph extraction), and it would be great to see a clearer side-by-side runtime comparison with BUDDY [1], including preprocessing (node-wise) and estimating structural features (link-wise).
> Your concern about scalability, particularly regarding feature propagation and subgraph extraction, is valid. We would like to clarify that MPLP, akin to ELPH and BUDDY, does not necessitate a separate subgraph extraction process. In Figure 4 of our paper, we present a direct runtime comparison between MPLP and ELPH. This comparison comprehensively includes all steps required for inference, such as preprocessing, to ensure an equitable evaluation. It's noteworthy that for this experiment, we even modified ELPH to match BUDDY's efficiency level at the inference stage. This involved running the message-passing phase once for ELPH and caching the results for subsequent use, thereby aligning its process with that of BUDDY.
>
>
>
> [1] Chamberlain, Benjamin Paul, et al. "Graph neural networks for link prediction with subgraph sketching." ICLR (2023).
>
> [2] Chen, Zhengdao, et al. "Can graph neural networks count substructures?." Advances in neural information processing systems 33 (2020): 10383-10395.
>
> [3] Wang, Xiyuan, Haotong Yang, and Muhan Zhang. "Neural Common Neighbor with Completion for Link Prediction." arXiv preprint arXiv:2302.00890 (2023).
>
> [4] Zhang, Muhan, and Yixin Chen. "Link prediction based on graph neural networks." Advances in neural information processing systems 31 (2018).
>
> [5] Yin, Haoteng, et al. "Algorithm and system co-design for efficient subgraph-based graph representation learning." VLDB (2022).
>
> [6] Hamilton, Will, Zhitao Ying, and Jure Leskovec. "Inductive representation learning on large graphs." Advances in neural information processing systems 30 (2017).

---

> ### Author Response · Authors · 2023-11-21
> **A kind reminder**
>
> Dear Reviewer UTLj,
>
> We sincerely appreciate the time and effort you have dedicated to reviewing our manuscript. Your insightful feedback has been invaluable, and we have taken careful steps to address each of your concerns in our revised submission.
>
> We believe that our responses and revisions have comprehensively addressed the points you raised. In light of these efforts, we kindly request you to consider raising your score. We are fully committed to ensuring that our work meets the highest standards and would be more than willing to address any additional concerns you may have.
>
> As the deadline for the rebuttal phase is fast approaching, we would be grateful for the opportunity to engage in further discussion at your earliest convenience.
>
> Thank you once again for your valuable contributions to enhancing the quality of our work.

---

> ### Comment · Reviewer_UTLj · 2023-11-23
> **Followup on Author Response**
>
> Thank the authors for their thorough responses and clarification of my concerns. However, there are several issues that need to be further addressed.
>
> - W3: Unfortunately, I couldn't find the evidence in the revised manuscript to support the claim that MPLP is scalable. Meanwhile, as the authors also pointed out, the density of the graph plays a vital role in the estimation quality, and its quantitative analysis is also lacking. It would be great to see a systematic quantitative analysis of MPLP estimating #(p,q) with different orders over graphs in different densities.
>
> - Q4: There are no quantitative results supporting the effectiveness of MPLP in estimating substructures such as triangles.
>
> - Q5: If GNNs in Eq. (4) and Eq. (8) are indeed vanilla GNNs, wouldn't MPLP be at best as efficient as ELPH, since the message passing cannot be precomputed as feature propagation?
>
> - Q6: Upon checking the description of selected baselines, I found the revised B.2 is still missing detailed model configuration and experimental setup, especially the network architecture, the structural features used, and the setting for their estimation.
>
> - Lastly, I would like to highlight how to interpret Fig. 4. and benchmark the efficiency of different models. From my understanding, the main difference between ELPH (BUDDY) and MPLP is the way structural features are estimated: the former adopts MinHash and HyperLogLog, and the latter uses simplified message passing with QO vectors. Thus, to verify the efficiency and superiority of MPLP, it is crucial to perform a side-by-side fair comparison in terms of node-wise preprocessing and pairwise estimation under a clear experimental setup and hyperparameters (feature types, dimension used for estimation, backbone models and their architectures, etc) that might affect final results. However, Fig. 4 does not quite convey such message.
>
> I appreciate the authors’ effort in improving the manuscripts. However, based on the issues listed above and concerns from other reviewers, the manuscript is not yet ready. I will maintain my initial rating and encourage the authors to incorporate the feedback from all reviewers for the latter revision.

---

> ### Author Response · Authors · 2023-11-23
> **2nd round respound to reviewer UTLj [1/3]**
>
> Dear Reviewers UTLj,
>
> Thank you for your rigorous review and thoughtful insights. Here are our responses to your queries:
>
> ### W3: Unfortunately, I couldn't find the evidence in the revised manuscript to support the claim that MPLP is scalable. Meanwhile, as the authors also pointed out, the density of the graph plays a vital role in the estimation quality, and its quantitative analysis is also lacking. It would be great to see a systematic quantitative analysis of MPLP estimating #(p,q)with different orders over graphs with different densities.
>
> The scalability of MPLP is primarily driven by its efficiency in estimating structural features. MPLP achieves the state-of-the-art performance (shown in Table 1,2) with comparable or even better efficiency (shown in Fig. 4) compared to other baseline models. In Fig. 4, MPLP is as fast as most of the advanced baseline models. Moreoever, MPLP (no feat), the version of MPLP only utilizes structural features, has much less number of parameters compared to other baseline models, achieving better performance than other baselines which are orders of magnitude larger.
>
> We appreciate that the reviewer points out the necessary of experimenting with graphs with different densities. MPLP, as the initial manuscript has pointed out, has its own limitaion when the density of graphs becomes high. To respond to the reviewer's query, we conducted experiments on graphs with different densities. We generate synthetic graphs with Stochastic Block Models with varying within-block edge densities. Then, similar to the Fig.5/Fig.8, we report the Mean Square Error of estimating #(p,q) with increasing average node degree. Due to the limited time, we report the results below and will incorporate it into the revised manuscript later as a figure.
>
> As the results shows, with a fixed computation budget ($F=1024$), the estimation error of MPLP will go higher when the graph becomes denser. This trend verifies what the Theorem 2 has pointed out about the estimation quality. While this experiment tested MPLP in the extreme cases, we want to emphasize that most of the graph in the real-world are highly sparse[7], which means that MPLP is widely adoptable in practice.
>
> | avg_degree | (1,2)  | (2,2)   | (1,inf) | (2,inf) | (1,1) |
> |------------|--------|---------|---------|---------|-------|
> | 21.784     | 14.07  | 125.52  | 15.69   | 520.15  | 0.41  |
> | 31.751     | 36.61  | 421.97  | 40.27   | 1742.72 | 0.88  |
> | 41.809     | 65.99  | 843.59  | 72.33   | 3508.17 | 1.55  |
> | 51.9       | 98.56  | 1335.65 | 108.78  | 5612.44 | 2.43  |
> | 61.839     | 127.03 | 1582.63 | 141.94  | 6704.63 | 3.47  |
> | 71.869     | 154.65 | 1670.11 | 175.21  | 7131.83 | 4.76  |
> | 81.833     | 181.97 | 1665.45 | 209.59  | 7174.42 | 6.16  |
> | 91.781     | 217.36 | 2095.07 | 253.73  | 9296.1  | 7.61  |
> | 101.758    | 233.34 | 1744.76 | 276.44  | 7659.97 | 9.74  |
> | 151.595    | 375.97 | 1806.18 | 481.65  | 8289.19 | 22.07 |
> | 201.736    | 528.93 | 1857.32 | 727.11  | 8856.65 | 40.14 |
> | 251.839    | 692.79 | 1890.7  | 1019.39 | 9355.0  | 63.65 |
>
> ### Q4: There are no quantitative results supporting the effectiveness of MPLP in estimating substructures such as triangles.
> We appreciate the reviewer pointing out the lack of quantitative results supporting the effectiveness of MPLP in estimating the number of triangles. While the scope of our study is to tackle the link prediction task and the triangle counting power is one aspect of MPLP, we want to respond to the reviewer's concern by conducting a quantitative analysis of MPLP's counting power. We take the triangle estimation component of MPLP out as a standalone model and apply it to count the triangles in the random graphs. We follow the same experimental setup from Sec 6.1 of [2], and compare MPLP with other baseline models by the median of MSE. The results are shown in the table below.
>
> | models       | Erdos-Renyi | Random Regular |
> |--------------|-------------|----------------|
> | GCN          | 8.27E-1     | 2.05           |
> | GIN          | 1.25E-1     | 4.74E-1        |
> | SAGE         | 1.48E-1     | 5.21E-1        |
> |--------------|-------------|----------------|
> | sGNN         | 1.13E-1     | 4.43E-1        |
> | 2-IGN        | 9.85E-1     | 5.96E-1        |
> | PPGN         | 2.51E-7     | 3.71E-5        |
> | LRP-1-3      | 2.49E-4     | 3.83E-4        |
> | Deep LRP-1-3 | 4.77E-5     | 5.16E-6        |
> |--------------|-------------|----------------|
> | **MPLP**     | 1.61E-4     | 3.70E-4        |
>
> As the results show, the triangle estimation component of MPLP can estimate the number of triangles in the graph with almost neglible error, similar to other more expressive models. Moreover, MPLP achieves this with a much lower computational cost, which is comparable to 1-WL GNNs like GCN, GIN and SAGE. It demonstrates MPLP's advantage of better efficiency over more complex GNNs like 2-IGN and PPGN.

---

> ### Author Response · Authors · 2023-11-23
> **2nd round respound to reviewer UTLj [2/3]**
>
> ### Q5, If GNNs in Eq. (4) and Eq. (8) are indeed vanilla GNNs, wouldn't MPLP be at best as efficient as ELPH, since the message passing cannot be precomputed as feature propagation?
> We appreciate the author pointing out the efficiency of MPLP equivalent to ELPH. Theorectically, the efficiency of ELPH and MPLP is comparable. However, in practice, we find that ELPH is not only inferior compared to MPLP in terms of estimation accuracy, but also much slower in terms of inference. Our initial manuscript has discussed this phenomenon in the right figure of Fig. 8. It shows that ELPH needs to take much higher computation walltime to perform their subgraph sketching operation, almost exponential to the number of node signature dimension $F$. After careful investigation, we find that the message passing of ELPH's sketching is less efficient compared to MPLP. We believe that this is due to two reasons:
>
> - Firstly, ELPH needs to take care of both sets of MinHash and Hyperloglog, while MPLP only requires one set of node signatures. This difference makes ELPH slightly slower than MPLP in terms of time complexity, and much memory hungry in terms of space complexity. We find that it can hit OOM on 32GB memory GPU when the node signature dimension surpasses 3000.
>
> - Secondly, the operations of MinHash and Hyperloglog is a max reduce operation, while MPLP is a simple sum reduce opeartion. This difference makes the operation of ELPH less efficient when running on GPU[8]. However, MPLP can enjoy the full GPU acceleration on the sparse-dense matrix multiplication.
>
> Therefore, ELPH shows less efficiency compared to MPLP in practice. In fact, to further boost the efficiency of MPLP, we can take a technique routine similar to BUDDY, which precompute the node representation/signatures before the model tranining. However, this may compromise the expressiveness of MPLP. But after all it also gives an evidence that MPLP is a flexible framework and there are a lot of opportunities to further improve it in terms of both efficiency and performance.
>
>
> ### Q6 Upon checking the description of selected baselines, I found the revised B.2 is still missing detailed model configuration and experimental setup, especially the network architecture, the structural features used, and the setting for their estimation.
> We appreciate the reviewer's suggestion about the description of model configurations. We have detailed the baseline model selection process in the revised B.2 to reflect what we have done. In general, we conduct a thorough hyperparamter tuning for each of our baseline methods to ensure a fair comparison. Besides, we select ELPH over BUDDY, and NCNC over NCN for the sake of higher capability of the models compared to their variants.

---

> ### Author Response · Authors · 2023-11-23
> **2nd round respound to reviewer UTLj [3/3]**
>
> ### Lastly, I would like to highlight how to interpret Fig. 4. and benchmark the efficiency of different models. From my understanding, the main difference between ELPH (BUDDY) and MPLP is the way structural features are estimated: the former adopts MinHash and HyperLogLog, and the latter uses simplified message passing with QO vectors. Thus, to verify the efficiency and superiority of MPLP, it is crucial to perform a side-by-side fair comparison in terms of node-wise preprocessing and pairwise estimation under a clear experimental setup and hyperparameters (feature types, dimension used for estimation, backbone models and their architectures, etc) that might affect final results. However, Fig. 4 does not quite convey such message.
> We appreciate the reviewer's highlight about our efficiency evaluation. In our initial manuscript, we have detailed the evaluation process between ELPH and MPLP in Appendix B.3. The walltime of our evalution includes all the node-wise preprocessing and the pairwise estimation. More precisely, for both ELPH and MPLP, we perform the message passing once on the entire graph to cache the node signatures (QO vectors for MPLP and MinHash/Hyperloglog for ELPH) and the node representation. Then, we perform the pairwise estimation on the test set based on the precomputed node signatures and node representation. This evaluation process is indeed fair and side-by-side, because it includes all necessary computation for one epoch of inference.
>
> As the reviewer points out, the efficiency of ELPH should be comparable to MPLP. However, as we have discussed in the response to Q5, ELPH is running slower in practice compared to MPLP. Moreover, MPLP, with better practical efficiency, can achieve better link prediction performance by introducing techniques like **Norm rescaling** to enable weighted node label counting. This has demonstrated the superiorty of MPLP in terms of both efficiency and performance compared to ELPH.
>
>
> [7] Fey, Matthias, and Jan Eric Lenssen. Fast graph representation learning with PyTorch Geometric.
>
> [8] Yang, Carl, Aydın Buluç, and John D. Owens. Design principles for sparse matrix multiplication on the gpu.
>
> -------
> Thank you for taking the time to review our work. We appreciate your feedback, which have greatly improved our manuscript. We have prepared a thorough response to address your concerns and hope it can address all of your questions. In light of this, we hope you consider raising your score.

---

### Author Response · Authors · 2023-11-23
**Response to the evaluation protocol of MPLP**

Dear AC and Reviewing Team,

We deeply appreciate the thorough reviews and constructive feedback. The suggestions and comments have been instrumental in improving the quality of our work. We address common concerns about our study raised by reviewers:

### The lack of experiments on other OGBL datasets
While OGBL[1] datasets becomes popular on benchmarking the link prediction methods, we argue that it is not a one-for-all standard for a comprehensive evaluation.

- Firstly, a recent study[2] found that OGBL-DDI may not be a suitable benchmark for the link prediction task. The authors of [2] found out that there exists a poor relationship between the validation and testing performance. This observation extends to recent link prediction models like Neo-GNN, NCNC, and ELPH, which we include as the baselines. The gap between validation and testing performance makes the hyperparmeter tuning on OGBL-DDI diffucult, and causes problems in reproducing the reported results. More details and discussion can be found in Appendices C and D in [2].

- Secondly, considering graph is a ubiqutous graph structure, OGBL datasets only cover a limited scope of domains. However, our benchmarks come from a more diverse selection of domains, including collaboration networks (NS,CS,Physics,Collab), transportation networks (USAir), electrical networks (Power), biological networks (Yeast,E.coli), internet networks (PB,Router), product networks (Computers,Photo), and neural networks (C.ele). We comprehensively evaluate MPLP and other baseline models on these datasets with a consistent experimental setup, and the results show that MPLP achieves state-of-the-art performance on most of them (12 out of 13). We believe that the evaluation of these diverse datasets is sufficient to demonstrate the effectiveness of MPLP.

- Thirdly, we acknowledge the limitation (Appendix F) in MPLP’s performance in estimating structural features in denser graphs (theoretically discussed in Theorem 2). However, the graph in the real world tends to be sparse [3], which indicates the wide applicability of MPLP in practice.


|            | DDI    | PPA   | Collab | Citataion2 |
|------------|--------|-------|--------|------------|
| Avg Degree | 312.84 | 52.62 | 5.45   | 10.44      |
|            |        |       |        |            |
| Histogram  | of     | #     | 2-hop  | neighbors  |
| [0,1)      | 0.0    | 0.0   | 42.5   | 0.0        |
| [1,10)     | 0.0    | 0.4   | 7.5    | 3.1        |
| [10,50)    | 0.2    | 1.4   | 24.3   | 9.9        |
| [50,100)   | 0.0    | 1.5   | 11.7   | 8.3        |
| [100,500)  | 0.6    | 9.4   | 12.7   | 31.0       |
| [500,1000) | 3.3    | 11.4  | 0.3    | 15.3       |
| [1000,inf) | 95.8   | 75.8  | 0.3    | 32.4       |


[1] Hu, Weihua, et al. Open graph benchmark: Datasets for machine learning on graphs. Neurips 2020.

[2] Li, Juanhui, et al. Evaluating Graph Neural Networks for Link Prediction: Current Pitfalls and New Benchmarking. Neurips 2023.

[3] Watts, Duncan J., and Steven H. Strogatz. Collective dynamics of ‘small-world’networks. Nature 1998.

-----

We hope that this response addresses the reviewers' concerns over the evaluation protocol of MPLP, and we eagerly await any further feedback you may have.

---

### Meta-Review · Area_Chair_Hbfc · 2023-12-09

**Metareview:**

In this paper, the authors studied the potential of message-passing GNNs to encapsulate joint structural features of graphs. Specifically, they introduced Message Passing Link Predictor (MPLP), a novel link prediction model that taps into quasi-orthogonal vectors to estimate link-level structural features while preserving the node-level complexities. The experimental results show the effectiveness of the proposed method.

The paper is well-written and easy to follow. However, the reviewers have many concerns about the scalable of the method, missing experiments on some OGBL datasets, lack of technical details, and quantitative results. Even though the authors tried to respond during rebuttal, the reviewers were still not satisfied with the answers.

**Justification For Why Not Higher Score:**

There are many concerns about the paper that need to be addressed in further study.

**Justification For Why Not Lower Score:**

N/A

---

### Decision · Program_Chairs · 2024-01-16

Reject